# Hydrogel crosslinking modulates macrophages, fibroblasts, and their communication, during wound healing

Sergei Butenko [1,2,16], Raji R. Nagalla[1,16], Christian F. Guerrero-Juarez [3], Francesco Palomba[1], Li-Mor David[1], Ronald Q. Nguyen [1], Denise Gay [2], Axel A. Almet[4,5], Michelle A. Digman [1,6], Qing Nie[4,5,7], Philip O. Scumpia [8,9,10], Maksim V. Plikus [2,4,11] & Wendy F. Liu [1,12,13,14,15] ✉

Biomaterial wound dressings, such as hydrogels, interact with host cells to regulate tissue repair. This study investigates how crosslinking of gelatin-based hydrogels influences immune and stromal cell behavior and wound healing in female mice. We observe that softer, lightly crosslinked hydrogels promote greater cellular infiltration and result in smaller scars compared to stiffer, heavily crosslinked hydrogels. Using single-cell RNA sequencing, we further show that heavily crosslinked hydrogels increase inflammation and lead to the formation of a distinct macrophage subpopulation exhibiting signs of oxidative activity and cell fusion. Conversely, lightly crosslinked hydrogels are more readily taken up by macrophages and integrated within the tissue. The physical properties differentially affect macrophage and fibroblast inter-actions, with heavily crosslinked hydrogels promoting pro-fibrotic fibroblast activity that drives macrophage fusion through RANKL signaling. These find-ings suggest that tuning the physical properties of hydrogels can guide cellular responses and improve healing, offering insights for designing better bioma-terials for wound treatment.

Impaired wound healing remains a global challenge, causing skin scarring or failed healing post-injury or surgery[1]. Biomaterial wound dressings offer a promising solution, modulating host immune and stromal cell activities to enhance healing[2,3]. Macrophages (Mφ) are key regulators of early inflammatory and later reparative stages of healing, clearing damaged tissue, resolving inflammation, and promoting repair. Proper and timely activation is crucial to prevent chronic wounds or fibrosis from excessive inflammation or repair[4]. Fibroblasts contract the wound and deposit new extracellular matrix (ECM) to form a collagen-rich scar[5]. Biomaterial dressings interact with host

[1]Department of Biomedical Engineering, University of California Irvine, Irvine, CA, USA. [2]Department of Developmental and Cell Biology, University of California, Irvine, Irvine, CA, USA. [3]Carle Illinois College of Medicine, University of Illinois at Urbana-Champaign, Urbana-Champaign, IL, USA. [4]NSF-Simons Center for Multiscale Cell Fate Research, University of California, Irvine, Irvine, CA, USA. [5]Department of Mathematics, University of California, Irvine, Irvine, CA, USA. [6]Laboratory of Fluorescence Dynamics, The Henry Samueli School of Engineering, University of California, Irvine, CA, USA. [7]Center for Complex Biological Systems, University of California Irvine, Irvine, CA, USA. [8]Division of Dermatology, Department of Medicine, David Geffen School of Medicine, University of California, Los Angeles, Los Angeles, CA, USA. [9]Jonsson Comprehensive Cancer Center, David Geffen School of Medicine, University of California, Los Angeles, Los Angeles, CA, USA. [10]Department of Dermatology, Veterans Administration Greater Los Angeles Healthcare System, Los Angeles, CA, USA. [11]Sue and Bill Gross Stem Cell Research Center, University of California, Irvine, Irvine, CA, USA. [12]UCI Edwards Lifesciences Foundation Cardio-vascular Innovation and Research Center, University of California Irvine, Irvine, CA, USA. [13]Institute for Immunology, University of California, Irvine, Irvine, CA, USA. [14]Molecular Biology and Biochemistry, University of California, Irvine, Irvine, CA, USA. [15]Department of Chemical and Biomolecular Engineering, University of California Irvine, Irvine, CA, USA. [16]These authors contributed equally: Sergei Butenko, Raji R. Nagalla. ✉e-mail: wendy.liu@uci.edu

cells in the wound, altering their environment and can be used to manipulate healing[3,6]. Some studies show biomaterials can promote skin regeneration via immune cell mechanisms[3,6,7]. Understanding how biomaterial properties regulate cell functions is vital for designing wound dressings that support repair while reducing scar.

Biophysical properties of hydrogel-based biomaterials can be tuned by modifying concentration, crosslinking, or gelation conditions[8,9], all of which modulate cell function. We and others have found that increased hydrogel crosslinking or stiffness enhances Mϕ inflammatory responses[10–12]. Stiffer environments also promote myofibroblast activation[7,13,14], and implantation of stiffer materials generally triggers inflammation and fibrotic scarring[3,15,16]. Crosslinking natural hydrogels such as collagen or gelatin not only changes their stiffness, but also degradability and host cell infiltration. However, the impact of biomaterial crosslinking on cellular interactions and molecular signaling in skin wound healing remains unclear.

Advances in single-cell RNA-sequencing (scRNA-seq) have revolutionized the study of host cell responses to biomaterials and wound healing. A recent study showed that fiber alignment in electrospun scaffolds shifted the response from innate to adaptive immunity, leading to hair follicle regeneration in small wounds[17]. Another study identified specific subpopulations of tissue-resident Mϕs as primary sources of biomaterial degradation[18]. Additionally, scRNA-seq has been used to create a cellular response atlas to biomaterials in a mouse wound model[19], reconstructing signaling networks around implanted biomaterials and identifying pathways associated with wound healing. These studies offer insights into wound healing dynamics, particularly the interplay between immune and mesenchymal cells involved in tissue repair.

Here, we use scRNA-seq to investigate the effects of varying gelatin methacrylate (GelMA) crosslinking on wound cell behavior. We compare soft, lightly crosslinked GelMA (lo-GelMA, 3 kPa) with stiffer, highly crosslinked GelMA (hi-GelMA, 150 kPa) applied to full-thickness murine skin wounds. Our histological and scRNA-seq analyses reveal distinct cellular responses: hi-GelMA induces enhanced inflammation and a foreign body-like response, while lo-GelMA promotes cellular integration and phagocytic removal of the biomaterial. In the hi-GelMA condition, Mϕ phenotype is skewed towards pro-inflammatory activation, with a subpopulation bordering the hydrogels generating a highly oxidative environment. Enhanced inflammation with hi-GelMA is accompanied by elevated pro-fibrotic fibroblast activity. Analysis of the wound transcriptome's cell-cell communication network identifies differential Mϕ-fibroblast bidirectional signaling, shifting the balance toward pro-inflammatory and pro-fibrotic activation in hi-GelMA compared to lo-GelMA. These findings suggest a crucial link between hydrogel crosslinking and cell-material interactions, emphasizing the importance of understanding these dynamics in biomaterial design for wound healing.

## Results

### GelMA crosslinking affects scar size and tissue architecture
To assess the effect of GelMA crosslinking on scarring and wound architecture, GelMA was applied to 5 mm full-thickness dorsal skin wounds. Lightly crosslinked hydrogels (lo-GelMA) were created using 365 nm light for 1 min, and heavily crosslinked hydrogels (hi-GelMA) were created with 5 min light exposure. All wounds were covered with Tegaderm and control wounds (Sham) were left untreated and covered only with Tegaderm (Fig. 1a). Wound tissue was collected for histology at post-wounding day (PWD) 5, 10, and 30. At PWD30, both lo- and hi-GelMA tended to result in smaller scar sizes compared to Sham; however, lo-GelMA showed a significant reduction in scar area (Fig. 1b). H&E staining of wounds at PWD5 and 10 showed that at PWD5, lo-GelMA had variously sized cellular infiltration channels, integrating well with the dermis, as fibrohistiocytic, endothelial, and other immune cells infiltrated the scaffold from the wound's edges and depth (Fig. 1c; Supplementary Fig. 1). In contrast, hi-GelMA showed

minimal cell infiltration and the onset of biomaterial extrusion caused by a stronger foreign body response (FBR). Furthermore, re-epithelialization over lo-GelMA at PWD5 covered 60-100% of the wound, while hi-GelMA covered only 2-5% (Fig. 1c), consistent with prior studies on porous scaffolds[20]. Aggregates on the periphery of hi-GelMA indicated poor biocompatibility, likely from unfavorable hi-GelMA-periwound interactions. Cross-sectional area of lo-GelMA ranged from 1–5 mm², while hi-GelMA measured 4–7 mm² (Fig. 1c), suggesting greater lo-GelMA degradation. By PWD10, lo-GelMA's cross-sectional area was 0–1 mm², while hi-GelMA remained unincorporated, causing re-epithelialization under the gel and eventual extrusion. Overall, lo- and hi-GelMA response appeared to vary in biocompatibility and cellular integration, and thus we hypothesized that the hydrogel and its extent of crosslinking regulates scarring through modulation of immune and stromal cells.

### GelMA crosslinking modulates early wound composition
We used scRNA-seq to investigate the impact of GelMA on wound composition at PWD5 (Fig. 1a). Wound tissues from five mice per treatment were pooled, enzymatically dissociated, and processed for RNA library preparation using 10X Chromium V3.1. Sequencing on NovaSeq yielded approximately 10,000 cells per sample at a depth of 50,000 reads per cell. Cell Ranger was used for data alignment, and analysis was performed in Seurat. Following quality control, the samples contained: 5400 cells for Sham control, 6579 for lo-GelMA, and 5007 for hi-GelMA. Louvain clustering revealed nine distinct cell clusters, visualized through UMAP (Fig. 1d) and differential gene signatures and canonical markers determined cluster identities (Fig. 1e). Fibroblasts and Mϕs dominated the wound populations (Fig. 1f, g). Fibroblasts averaged 57.2% of the cells, with lo-GelMA having the highest percentage (66.1%) compared to hi-GelMA (50.1%) and Sham (55.3%) (Fig. 1f, g). Mϕs made up 20.9% of total cells on average and was highest in Sham (25.1%), and lower in hi-GelMA (20.2%) and lo-GelMA (17.5%). Neutrophils, averaging 14.5%, almost doubled in hi-GelMA (22.5%) versus lo-GelMA (9.5%) and Sham (11.6%), suggesting elevated inflammation in hi-GelMA.

Biomaterial conditions also affected populations that were present in lower abundance. The T cell and natural killer (T+NK) cell cluster was more prevalent in Sham (1.5%) and hi-GelMA (1.9%) compared to lo-GelMA (0.7%). Langerhans cells (LC) and mast cells were more abundant in hi-GelMA (1.2% LC, 0.8% mast cells) versus lo-GelMA (0.8% LC, 0.3% mast cells) or Sham (0.9% LC, 0.4% mast cells). Keratinocytes were at low levels in Sham and lo-GelMA (1.5% each) but nearly absent in hi-GelMA (0.1%). Lack of re-epithelialization was likely due to the Tegaderm dressing used in all cases and worsened with hi-GelMA. Dendritic cell (DC) percentages remained similar across treatments, averaging 0.9%. Fibrocytes (Mϕ/Fib), marked by a mix of fibroblast and Mϕ markers[21], averaged 2.6% across conditions. These results show that GelMA and crosslinking degree influence wound cell infiltration, with hi-GelMA inducing more inflammatory cells (neutrophils), while lo-GelMA exhibiting advanced healing (fibroblasts, keratinocytes) five days post-treatment.

### GelMA crosslinking modulates macrophage subpopulations
To explore the crosslinking effect on Mϕ immunomodulation, we analyzed the $Cd68^{high}$/ $Mrc1^{high}$ population by subclustering it into six functional subpopulations (M1-6) based on differential gene signatures compared to canonical markers (Fig. 2a) and Gene Ontology (GO) characterization (Fig. 2b–d; Supplementary Figs. 2–4). Comparing hi- and lo-GelMA, subpopulations M1 and M4 were upregulated in hi-GelMA, while M5 and partially M3 were upregulated in lo-GelMA. M2 and M6 were largely unaffected by GelMA crosslinking, but M6 was reduced in both GelMA conditions compared to Sham.

Clusters upregulated by hi-GelMA−M1 and M4−displayed pro-inflammatory signatures, with M4 also showing a distinct oxidative

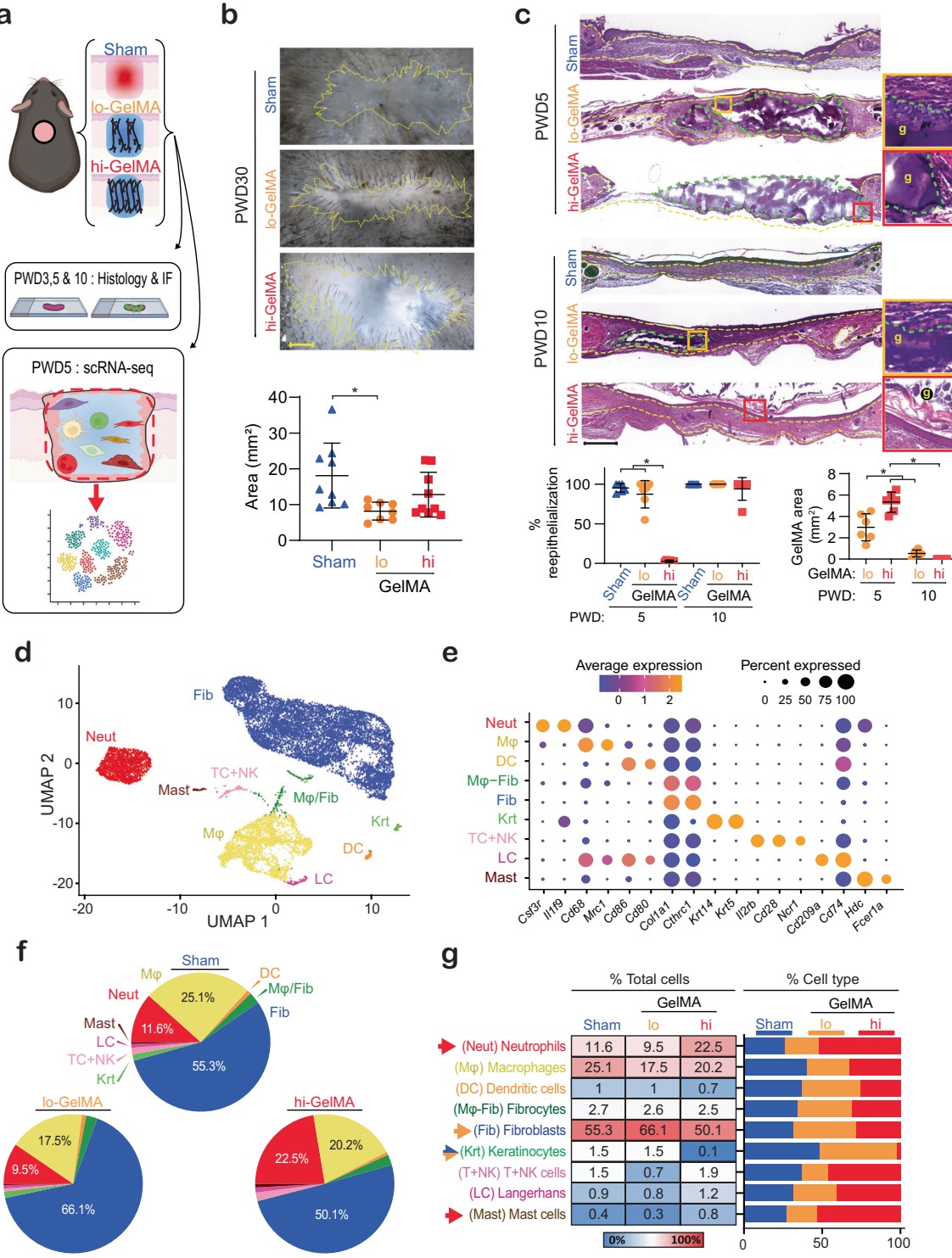

**Fig. 1 | Crosslinking of GelMA hydrogel dressing modulates scarring, healing dynamics and cell infiltrate of full-thickness murine skin wounds. a** Schematic of wounding studies. **b** Images of fixed, whole mounted wounds treated with lo- or hi-GelMA, or Sham (Tegaderm only), at PWD30. Scar outlined with yellow dashed line. *n* = 9 (mice). Scale bar: 1 mm. *p* = 0.01 (one-way ANOVA, Tukey's HSD). **c** Images of H&E stained PWD5 and 10 showing incorporation and degradation of lo-GelMA versus inflammation with cell aggregation and extrusion of hi-GelMA. Yellow and red boxes indicate regions that have been shown at higher magnification to the right. Yellow dashed line: epithelium-dermis and dermis-hypodermis border. Green dashed line: GelMA-dermis border. g: GelMA. Scale bar: 200 μm. Quantification of epithelial migration from wound margins defined as a percentage of re-epithelialization, and the degradation of GelMA is measured as the area of GelMA. *n* = 6 (mice). Re-epithelialization: *p* < 0.0001; GelMA area: PWD5 lo/hi: *p* = 0.0003,

lo-GelMA PWD5/10: *p* = 0.0002, hi-GelMA PWD5/10: *p* < 0.0001 (one-way ANOVA, Tukey's HSD) **d** UMAP dimensional reduction representing cells categorized into nine main clusters with each cell color-coded based on its cell type. **e** Dot plot of total cells showing expression of canonical markers of each cell type. Dot size corresponds to the proportion of cells within the group expressing each gene, and color correspond to expression level. **f** Pie chart and (**g**) table along with bar plots showing cell populations of each cluster presented by percentages of total sequenced cells or percentages of specific cell type across treatments. Arrows indicate cell populations of interest that exhibit differential percentages between treatments. Blue shows populations enriched in Sham, orange shows populations enriched in lo-GelMA, and red shows populations enriched in hi-GelMA. All data presented as mean ± SD. Source data are provided as a Source Data file.

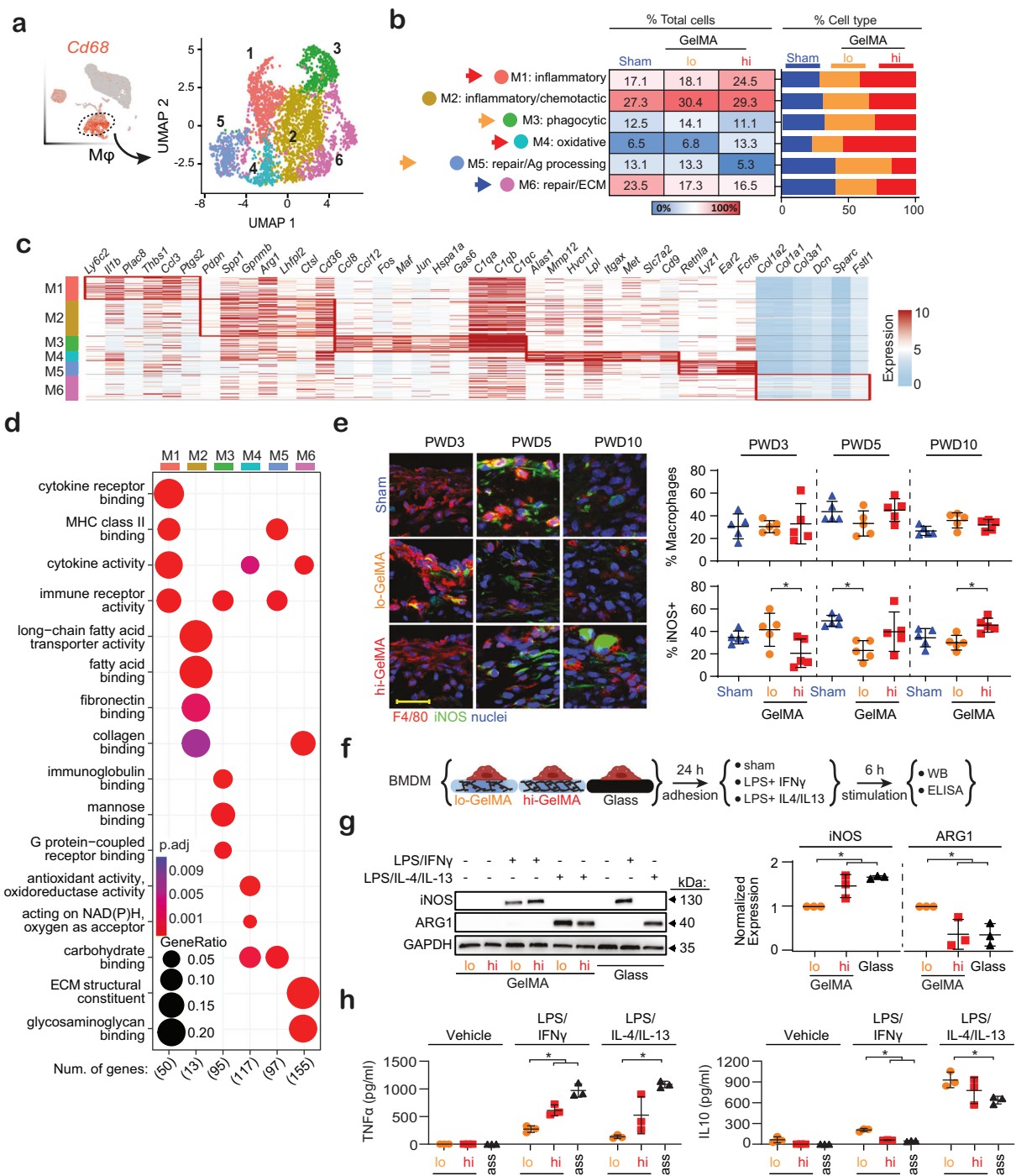

**Fig. 2 | Crosslinking of GelMA hydrogel dressing modulates Mφ function.**
**a** UMAP representation of Mφ cells from lo- or hi-GelMA and Sham wounds were categorized into six subpopulations represented by UMAP, with each cell color-coded by its associated cell cluster. **b** Table and bar plots showing Mφ populations of each cluster presented by percentages of total sequenced cells and percentages of specific cell types across treatments. Arrows indicate cell populations of interest that exhibit differential percentages between treatments. Color indicates the enriched treatment: blue (Sham), orange (lo-GelMA), and red (hi-GelMA). **c** Heatmap showing differentially expressed genes serving as phenotypic markers. **d** Dot plot of the enriched GO biological processes of highly expressed genes in each subpopulation. Dot size corresponds to the proportion of cells within the group expressing each gene, and dot color correspond to GO enrichment p-value. **e** Immunohistochemistry of lo- or hi-GelMA and Sham treated wounds at PWD3, 5 and 10, stained for F4/80 (Mφ marker) and iNOS (inflammatory marker). For full

image see Supplementary Fig. 6a. *n* = 5 (mice). Scale bar: 50 μm. PWD3: *p* = 0.03; PWD5: *p* = 0.01; PWD10: *p* = 0.01 (one-way ANOVA, Tukey's HSD). **f** Schematic of in vitro studies. **g** Immunoblots of cell lysate from BMDM probed for ARG1 (healing) and iNOS (inflammatory) expression, along with quantification across *n* = 3 (BMDM harvests from individual mice). iNOS: lo/hi: *p* = 0.02, lo/G: *p* = 0.004; ARG1: lo/hi: *p* = 0.04, lo/G: *p* = 0.04 (one-way ANOVA, Tukey's HSD). **h** ELISA of TNF-α and IL-10 secretion from BMDM, with quantification across *n* = 3 (BMDM harvests from individual mice). TNF-α: LPS/IFN-γ: lo/hi: *p* = 0.01, lo/G: *p* = 0.0003, LPS/IL-4/13: lo/G: *p* = 0.002; IL-10: LPS/IFN-γ: lo/hi: *p* = 0.0001, lo/G: *p* = 0.0001, LPS/IL-4/13: lo/G: *p* = 0.04 (one-way ANOVA, Tukey's HSD). All data presented as mean ± SD. Source data are provided as a Source Data file. Figure 2/panel f Created with BioRender.com released under a Creative Commons Attribution-NonCommercial-NoDerivs 4.0 International license (https://creativecommons.org/licenses/by-nc-nd/4.0/deed.en).

signature. M1 constituted 18.1% of Mφs in lo-GelMA and 17.1% in Sham but rose to 24.5% in hi-GelMA (Fig. 2b). This cluster included monocyte markers *Ly6c* and *Ccr2* and early inflammation markers *Il1b*, *Ccl3*, and *Ptgs2* (Fig. 2c; Supplementary Fig. 2a). GO analysis showed cytokine activity together with integrin and collagen binding, suggesting inflammatory activity coupled with extravasation or adhesion to the extracellular matrix (Fig. 2d; Supplementary Fig. 2a). Differential expression analysis showed that M1 in hi-GelMA were more active in early-inflammatory genes such as alarmins (*S100a8*, *S100a9*) compared to lo-GelMA (Supplementary Fig. 2a). Similarly, M1 in hi-GelMA, but not lo-GelMA, had increased antimicrobial defense genes (*S100a8*, *S100a9*, *S100a11*, *Ly6c*, *Ifitm1*, and *Chil3*) compared to Sham. M4 exhibited a pro-inflammatory signature with elevated *Il1b*, catabolic activity (*Mmp12*), and oxidative activity (*Hvcn1*) (Fig. 2c; Supplementary Fig. 2b). GO analysis also showed high oxidative activity (Fig. 2d; Supplementary Fig. 2b). M4 presence nearly doubled in hi-GelMA (13.3%) versus lo-GelMA (6.8%) and Sham (6.5%) (Fig. 2b), suggesting hydrogel crosslinking significantly regulates M4 levels. Notably, neutrophils were enriched in hi-GelMA (Fig. 1f, g), with increased expression of inflammatory genes and GO terms related to neutrophil extravasation, activation, and degranulation (Supplementary Fig. 5a–c). Together, these data suggest that hi-GelMA induces more inflammation than lo-GelMA or Sham.

M5 and M3 markers corresponded to pro-healing and phagocytic signatures, respectively, and were more prevalent in lo-GelMA compared to hi-GelMA (M5: 13.3% in lo-GelMA versus 5.3% in hi-GelMA; M3: 14.1% in lo-GelMA versus 11.1% in hi-GelMA) (Fig. 2b–d; Supplementary Fig. 3a, b). M5 showed pro-healing and antigen processing activity, expressing *Ear2* and *Retnla*, which are linked to healing, and GO terms including MHC class II and antigen processing (Fig. 2c, d; Supplementary Fig. 3b). Differential gene analysis revealed higher activity of the pro-healing genes *Tnfaip6*[22] in lo-GelMA versus hi-GelMA or Sham, and *Lgals1*[23] in lo-GelMA versus Sham (Supplementary Fig. 3b). M3 displayed complement, phagocytic, and antigen processing activities, with GO terms including immunoglobulin and carbohydrate binding, and expression of phagocytic marker *Mertk*, and phagosome maturation marker *Lamp1* (Fig. 2c, d; Supplementary Fig. 3a).

Percentages of M2 and M6 were unaffected by GelMA crosslinking. M2, the largest Mφ subpopulation, averaged 28% across all treatments and exhibited pro-inflammatory, chemotactic, phagocytic, and lipid-processing activities (Fig. 2b–d). M2 also expressed genes linked to homeostatic tissue-resident Mφ functions including phagocytosis and alternative activation (type 2), such as *Cd36*, *Ctsl*, and *Arg1* (Fig. 2c). GO terms included fatty acid binding and transport as well as ECM binding (Fig. 2d; Supplementary Fig. 4a). In hi-GelMA, M2 had higher expression of inflammatory genes *S100a8* and *S100a9* compared to lo-GelMA (Supplementary Fig. 4a). M6 exhibited reparative and collagenic activity, expressing genes supporting collagenous ECM (*Dcn*, *Col1a1*, *Col1a2*) and ECM-related GO terms (Fig. 2c, d; Supplementary Fig. 4b). M6 also expressed mechanoresponsive genes, such as *Yap1*, *Wwtr1* (TAZ), *Ptk2*, *Tln2*, *Syne2*, and *Pla2g4a* (Supplementary Fig. 4b). M6 presence was higher in Sham (23.5%) versus lo-GelMA (17.3%) and hi-GelMA (16.5%) (Fig. 2b), suggesting that GelMA reduces mechanosensitive and collagenic Mφs, potentially contributing to reduced scarring. M6 in hi-GelMA expressed higher levels of inflammatory genes (*S100a8*, *S100a9*, *Il1b*) compared to lo-GelMA, suggesting type 1 activation due to higher crosslinking (Supplementary Fig. 4b). In contrast, M6 in lo-GelMA expressed more ECM repair genes such as *Nov* and *Sparc*[24,25], compared to hi-GelMA. Thus, although M6 percentages were similar in lo- and hi-GelMA, the extent of activation was modulated by crosslinking.

## hi-GelMA enhances macrophage inflammation

To further examine the effect of GelMA on Mφs, wounds were collected at PWD3, 5, and 10, and Mφs were assessed for inflammatory activation via immunohistochemistry (Fig. 2e; Supplementary Fig. 6a). Mφ (marked by F4/80) rose marginally at PWD5 in hi-GelMA versus lo-GelMA. The percentage of Mφs expressing the inflammatory marker iNOS was significantly higher in lo-GelMA at PWD3 but decreased by PWD5, falling below Sham levels. In contrast, macrophage iNOS in hi-GelMA peaked at PWD10, significantly surpassing lo-GelMA, indicating that hi-GelMA sustains Mφ pro-inflammatory activation.

To directly investigate the effect of GelMA on Mφ polarization, we cultured bone marrow-derived Mφs (BMDM) on lo- or hi-GelMA, or fibronectin-coated glass (stiff control). BMDM on lo-GelMA appeared rounded with dendritic processes, while those on hi-GelMA and glass were flat (Supplementary Fig. 6c). After LPS stimulation, BMDM on lo-GelMA showed less inflammatory iNOS (with IFN-γ priming) and more ARG1 (with IL-4/13 priming) compared to hi-GelMA and glass (Fig. 2g; Supplementary Fig. 6b). BMDM supernatant showed lowered TNF-α and increased IL-10 cytokine secretion with lower GelMA crosslinking (Fig. 2h). These in vitro findings support a role for GelMA crosslinking in modulating Mφ activation, with higher crosslinking associated with increased type I inflammatory immune polarization.

## hi-GelMA enhances macrophage oxidative stress and glycolysis

M4-associated genes and GO terms showed pro-inflammatory, catabolic, and oxidative activity, with this cluster more prominent in hi-GelMA than lo-GelMA (Fig. 2b, d; Supplementary Fig. 2b). In this population, we observed that M4-expressed genes linked to oxidative stress—*Cat*, *Cybb*, *Sod1*, *Hif1a*, and *Ucp2*—were more pronounced in hi-GelMA versus lo-GelMA (Fig. 3a). Elevated oxidation may cause prolonged tissue damage and delay inflammation resolution, resulting in increased scarring. Additionally, M4 Mφs displayed high expression of key ECM proteases, *Mmp12* and *Mmp9*, which were also higher in hi-GelMA than lo-GelMA (Fig. 3b)[26]. We hypothesize that these Mφs directly respond to hi-GelMA and significantly contribute to increased inflammation and fibrosis.

To pinpoint the spatial location of M4 within the wound relative to the hydrogel, we identified *Cybb*, a highly expressed target encoding the Mφ key superoxide generating enzyme NADPH oxidase[27], in this Mφ subpopulation (Fig. 3c). Notably, M4 in hi-GelMA exhibited higher *Cybb* levels than in lo-GelMA. We then targeted *Cybb* and tracked its expression through RNA probe hybridization (RNAscope) on PWD5 histological sections (Fig. 3d). *Cybb*-expressing Mφs were dispersed and infiltrating into lo-GelMA scaffolds, but in contrast, *Cybb*-expressing Mφs densely clustered along the GelMA-dermis interface of hi-GelMA. This suggests that higher crosslinking in hi-GelMA, causes the accumulation of inflammatory, highly oxidative Mφs at the biomaterial-tissue boundary after these cells fail to migrate into the scaffold.

Enhanced oxidative activity and increased M4 presence in hi-GelMA, along with the greater numbers and pro-inflammatory activation of neutrophils, indicate ROS-enriched wounds. To evaluate the effect of GelMA crosslinking on wound ROS levels, we injected mice at PWD1, 2, and 3 with the ROS-responsive luminescent molecule luminol[28]. Wounds treated with hi-GelMA showed higher ROS levels on PWD1 than lo-GelMA (Fig. 3e). The signal was mostly detectable on PWD1 and decreased at PWD2 and 3, possibly due to diminishing early wound ROS sources like neutrophils. The higher neutrophil influx in hi-GelMA, and increased expression of ROS-related genes, may explain this stronger signal (Supplementary Fig. 5d). To assess the direct impact of GelMA crosslinking on Mφ ROS production, we cultured BMDM on GelMA of varying crosslinking or on fibronectin-coated glass (stiff control) and measured ROS levels after LPS stimulation. ROS production was higher in Mφs on hi-GelMA than lo-GelMA (Fig. 3g), consistent with the wound effect.

Since inflammatory activation and ROS release in Mφs are closely tied to glycolytic metabolism[4,29,30], we examined Mφ metabolism using

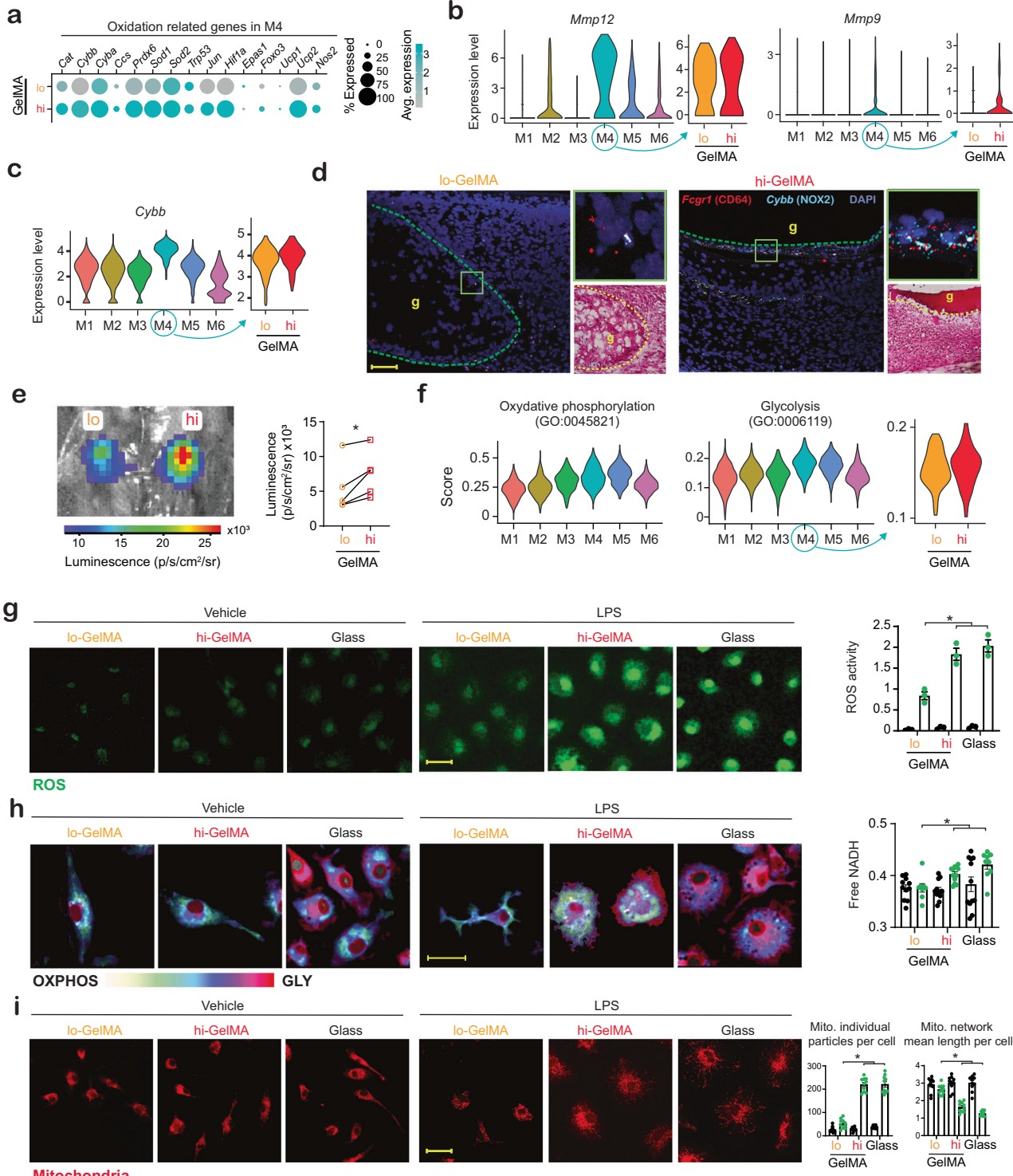

**Fig. 3 | GelMA crosslinking modulates Mφ during wound healing. a** Oxidation genes expressed by M4 in wounds with lo- and hi-GelMA. Dot size indicates the proportion of cells expressing each gene; dot color reflects expression levels. **b** Violin plots showing *Mmp12* and *Mmp9* expression across all Mφ (left) and comparing hi- versus lo-GelMA in M4 (right). **c** Violin plots of *Cybb* expression in all Mφ (left) and comparing hi- versus lo-GelMA in M4 (right). **d** RNAscope images of PWD5 wounds treated with lo versus hi-GelMA, stained for *Fcgr1*/CD64 (Mφ marker) and *Cybb*/Nox2 (oxidation gene). Green box: magnified view and H&E stain; green dashed line: GelMA-dermis border. g: GelMA. Scale bar: 50 μm. Representative images from *n* = 3 (wounds). **e** Wounds after a 100 mg/kg luminol injection, with luminescence quantified across *n* = 5 mice; *p* = 0.02 (two-sided paired t-test). Scale bar: 2 mm. **f** Signature scoring for oxidative phosphorylation (GO:0006119) and glycolysis (GO:0045821) in all Mφ and comparing hi- versus lo-GelMA in M4

Mφs (right). **g** BMDM seeded on hi- or lo-GelMA or glass for 24 h, stimulated with 1 μg/ml LPS for 16 h, stained with CellROX-green detecting ROS, with quantification. Scale bar: 25 μm, *n* = 3 (BMDM harvests from individual mice), *p* < 0.0001 (one-way ANOVA, Tukey's HSD). **h** FLIM images of BMDM seeded on hi- versus lo-GelMA or glass for 24 h, stimulated with 1 μg/ml LPS for 16h, with quantification (right). Scale bar: 15 μm. See Supplementary Fig. 7a for FLIM analysis. *n* = 9, 11, 12, 12, 12, and 11 for lo/V, lo/LPS, hi/V, hi/LPS, G/V, and G/LPS, respectively (cells from 3 mice). lo/hi: *p* = 0.02, lo/G: *p* = 0.0009 (one-way ANOVA, Tukey's HSD). **i** BMDM seeded on hi versus lo-GelMA or glass for 24 h, stimulated with LPS for 16 h, stained with MitoTracker Red CMXRos, with mitochondrial particles and network branches quantified. See Supplementary Fig. 7c for analysis. *n* = 10 (ROIs from 3 mouse BMDM culture). Scale bar: 25 μm. *p* < 0.0001 (one-way ANOVA, Tukey's HSD). All data presented as mean ± SD. Source data are provided as a Source Data file.

gene scoring (Ucell R package[31]) for glycolysis (GO:0045821) versus oxidative phosphorylation (GO:0006119). M4 showed a higher glycolytic score, while the pro-healing M5 had a higher oxidative phosphorylation score (Fig. 3g). The M4 glycolytic score was higher in hi-GelMA than lo-GelMA. To confirm GelMA crosslinking effects on Mφ metabolism in vitro, BMDM were seeded on GelMA with different crosslinking or glass, then stimulated with LPS and analyzed via fluorescence lifetime imaging microscopy (FLIM) to observe NADH/ NAD+ autofluorescence[32]. FLIM showed that BMDM on hi-GelMA exhibited a higher NADH/NAD+ ratio, indicating increased glycolysis compared to cells on lo-GelMA, which relied more on oxidative phosphorylation (Fig. 3h; Supplementary Fig. 7a). This suggests Mφ-hi-GelMA interactions shift metabolism towards glycolysis.

In wounds treated with hi-GelMA, Mφs expressed genes linked to mitochondrial dynamics (Supplementary Fig. 7b). Mitochondrial fission and fusion were analyzed by staining the mitochondrial network in BMDM cultured on GelMA with varying crosslinking or on glass, followed by LPS stimulation (Fig. 3i; Supplementary Fig. 7c)[33]. BMDM cultured on hi-GelMA showed increased mitochondrial fission with higher particle counts and shorter mean lengths per cell, characteristic of acute ROS environments[34]. These results suggest that hi-GelMA promotes acute ROS signaling in Mφs more than lo-GelMA, inducing Mφ adhesion, aggregation at the hydrogel surface, and elevated oxidative signaling.

## Modulation of macrophage phagocytosis by GelMA crosslinking

While M4 displayed catabolic activity, GO analyses showed that M3, which was enriched in lo-GelMA, exhibited phagocytic activity (Fig. 2b–d; Supplementary Fig. 2b, 3a). In M3, and to a lesser extent in M4, genes linked to phagocytic activity (*Lamp1, Rab7, Rab31, Trem2*) were more prominent in lo-GelMA than in hi-GelMA (Fig. 4a). To compare Mφ phagocytosis of hi- and lo-GelMA, labeled BMDM (Cell-Tracker, green) were seeded on hydrogels tagged with a pH-sensitive dye (pHrodo, red). After 12 h, BMDM showed significantly higher phagocytosis of lo-GelMA versus hi-GelMA (Fig. 4b). To explore how integrins might influence Mφ-GelMA interactions, we examined the expression profiles of integrins associated with adhesion and phagocytosis. We found that *Itgal, Itgax, Itgb2,* and *Itgav* were more upregulated in M4 compared to other subpopulations (Fig. 4c). BMDM seeded in vitro on GelMA for 24 hours showed increased the expression of *Itgal, Itgax, Itgb2, Itgav,* and to some extent *Itga5* (Fig. 4c). These integrins had similar expression levels on lo- and hi-GelMA, except *Itgav,* which was higher on lo-GelMA (Supplementary Fig. 8a).

Integrins form complement receptors to enhance phagocytosis: for example, complement receptors 3 and 4 (CR3 and CR4) are formed from CD18 (*Itgb2*) and CD11b (*Itgam*) or CD11c (*Itgax*) pairs[35]; LFA-1 complex forms from CD18 and CD11a (*Itgal*) pairs; *Itgav* (CD51) pairs with β1 and β3 subunits. To determine whether GelMA affects the expression of complement components relevant to CR3 and CR4 interactions, we analyzed wound scRNA-seq data and gene expression of BMDM cultured on GelMA (Fig. 4d). Results show upregulated *C3*, a key complement activation component, in M3 and M4 clusters, enhanced in lo-GelMA compared to hi-GelMA both in vivo and in vitro. While in vivo data show high expression of complement activator C1q components (*C1qa, C1qb, C1qc*) in M3, this upregulation was not observed in vitro.

To further investigate the impact of CR3 and CR4 integrin function on GelMA turnover, BMDM were seeded on lo-GelMA, which is more easily phagocytized, and treated with blocking antibodies against integrins CD18, CD11c, CD51, CD11a, and CD11b. These effects were compared to those of the less phagocytosed hi-GelMA and Cytochalasin-D (Cytoch.D, phagocytosis inhibitor) and isotype antibody-treated controls. Blocking CD18 (CR3 and CR4 components) and CD11c (CR4 component) reduced uptake, and their combination amplified this effect (Fig. 4e, f; Supplementary Fig. 8b). Treatments did

not affect BMDM adhesion (Supplementary Fig. 8c). Further analysis of hydrogel properties show lo-GelMA is softer (3 kPa) with higher porosity while hi-GelMA is stiffer (150 kPa) with lower porosity (Supplementary Fig. 9a, b). Mφ morphology aligned with phagocytic activity: phalloidin staining showed a dendritic shape of BMDM on lo-GelMA, which was reduced by blocking CD18 and CD11c, and flattened cells on hi-GelMA (Supplementary Fig. 9c). Together, these findings suggest that lo-GelMA promotes cellular extensions into the hydrogel pores and complement-dependent phagocytosis, while hi-GelMA promotes adhesion and limits phagocytosis.

## GelMA crosslinking modulates fibroblast subpopulations

We next studied the immunomodulatory effects of GelMA crosslinking on wound fibroblast phenotypes. *Col1a1*high/*Cthrc1*high fibroblasts segregated into six distinct functional subpopulations, which varied in their proportions across different wound treatments (Fig. 5a, b). Functional phenotypes were associated with differential gene signatures, canonical markers, and GO analysis (Fig. 5d, e). Subpopulations F1, F5, and F6 were more prevalent in hi-GelMA versus lo-GelMA, whereas F2, F3, and F4 increased in lo-GelMA relative to hi-GelMA (Fig. 5b).

In hi-GelMA, myofibroblast clusters (F1 and F6) were most upregulated, expressing genes related to contractile wound closure and scarring. F1, which averaged 15.5% of fibroblasts, was more prevalent in hi-GelMA (18.9%) and Sham (18.8%) versus lo-GelMA (11.8%). This cluster expressed *Col12a1* and *Cxcl12*, to some extent *Acta2* and *Tagln*, along with GO pathways involving ECM production, myofibroblast differentiation, and cytokine/ chemokine activity (Fig. 5d, e; Supplementary Fig. 10a). F6 was more frequent in hi-GelMA (14%) and Sham (11.6%) than lo-GelMA (6.3%), with myofibroblast markers *Acta2, Tagln, Col7a1,* and *Col15a1*, and GO terms of ECM structure and binding (Fig. 5d, e; Supplementary Fig. 10c). F6 also had higher proliferation markers including *Mki67* and G2/M phase genes (Supplementary Fig. 11c). Together, F1 and F6 made up 33% of fibroblasts in hi-GelMA, 31% in Sham, and 18% in lo-GelMA. Lastly, F5 was more abundant in hi-GelMA (11.4%) and Sham (12.4%) compared to lo-GelMA (8.7%). This angiogenic cluster expressed *Tek/Tie2, Igfbp5,* and *Clec3b*, with GO terms related to vasculogenesis (Fig. 5e; Supplementary Fig. 10b).

F2, F3, and to some extent F4, increased in lo- versus hi-GelMA. F2 showed the most significant rise, reaching 40.9% in lo-GelMA versus 29.8% in hi-GelMA and 35.6% in Sham (Fig. 5b). This cluster expressed chemotaxis-associated genes including *Ccl2, Cxcl1, Ccl7,* and *Cxcl10* (Fig. 5d), as well as GO terms related to chemokine, growth factor activity, and matrix adhesion (Fig. 5e). F3 also rose in lo-GelMA, at 16.4% versus around 12% in hi-GelMA and Sham. It expressed genes for collagen maturation (*Loxl2, Tnc, Cd44,* and *Col5a3*) (Fig. 5d) and GO terms of ECM organization and structure, which included genes such as *Cav1, Thy1,* and *Lgals3* that are known to negatively correlate with myofibroblast-like activity[36,37] (Supplementary Fig. 11b). F4 moderately increased to 15.9% in lo-GelMA versus 13.8% in hi-GelMA, expressing genes and GO terms related to proliferation including *Mki67*, tubulin, and histone binding (Fig. 5b; Supplementary Fig. 11c). F4 was distinct from F6, which also expressed proliferating genes, because it lacked ECM-related genes (Fig. 5d; Supplementary Fig. 11c). To summarize, clusters that were increased with lo-GelMA showed signatures of chemotaxis, proliferation, and collagen maturation, all demonstrating a progressing wound healing phenotype.

To assess the association between fibroblast subpopulations and ECM activity, we re-clustered fibroblasts with respect to ECM synthesis genes only, rather than the full gene set[38,39]. Five subpopulations were identified (Fig. 5c) including a subpopulation with high ECM synthesis gene expression (*Tgfbi, Ltbp2, Mfap4,* and *Spp*)[40–43], which closely aligned with the original pro-fibrotic F1 and F6 subpopulations. This group was twofold higher in hi-GelMA (22%) compared to lo-GelMA (11%). Using Picrosirius Red (PSR) staining, we evaluated collagen deposition and the fibrotic phenotype in the dermis (Fig. 5f;

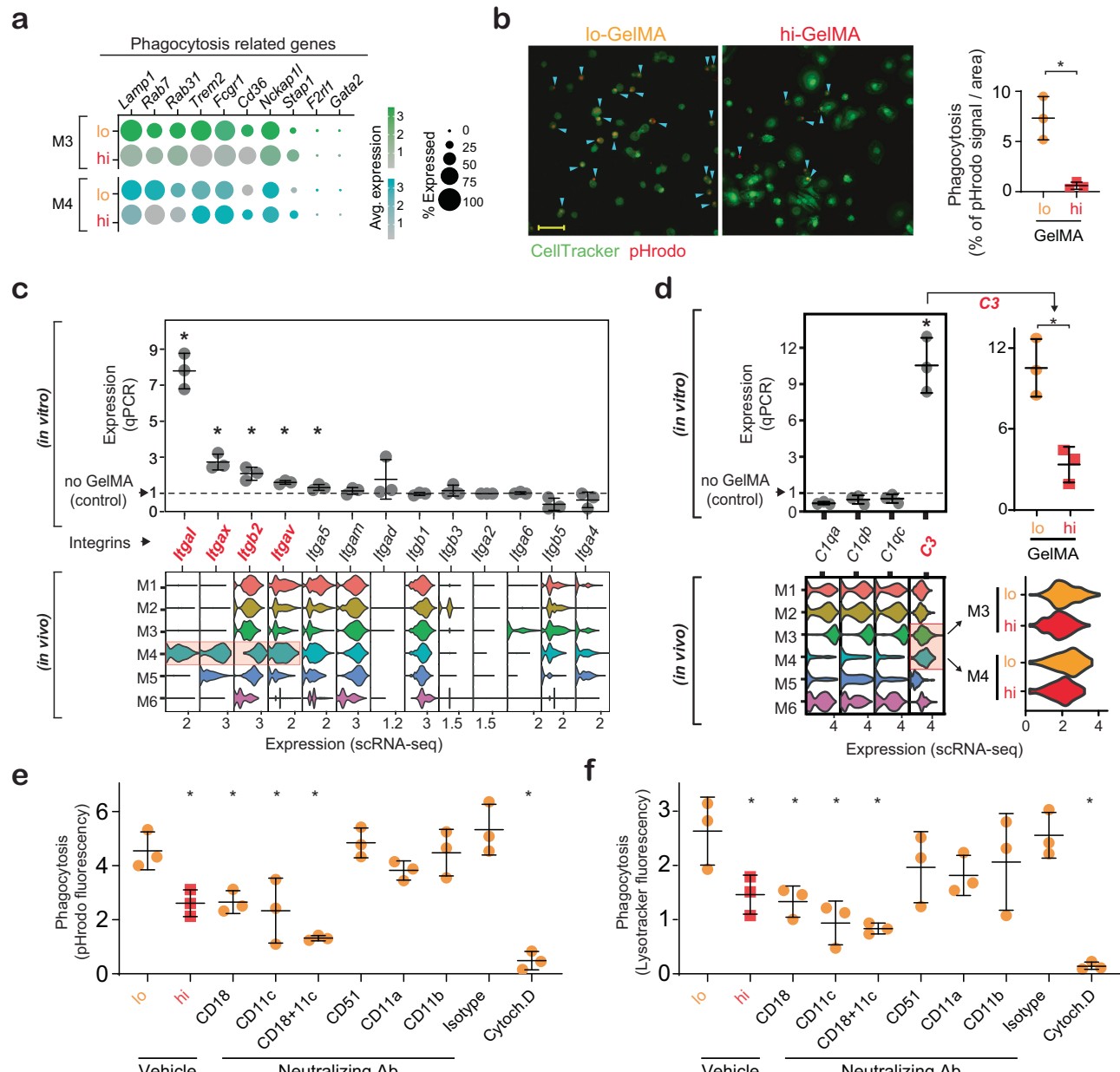

**Fig. 4 | Differential modulation of macrophage phagocytosis by GelMA crosslinking. a** Dot plot of phagocytosis- related genes expressed by M3 and M4 cells in lo- and hi-GelMA treated wounds. Dot size corresponds to the proportion of cells within the group expressing each transcript, and dot color correspond to the expression level. **b** Representative laser scanning microscopy images of labeled BMDM (green) seeded on pHrodo-labeled (red) hi- or lo-GelMA for 16h, along with quantification. Cyan arrowheads: phagocytosed GelMA. Scale bar: 25 μm. n = 3 (BMDM harvests from individual mice). p = 0.005 (t-test, two-sided). **c** Violin plot of integrin gene expression across in vivo Mφ subpopulations (bottom) and qPCR validation (top) of integrin gene expression in BMDM cultured on GelMA for 24h, compared to BMDM cultured without GelMA (control), n = 3 (BMDM harvests from individual mice). Left to right: p = 0.003, 0.002, 0.006, 0.005, 0.01(t-test, two-sided). **d** Violin plot of complement components gene expression across in vivo Mφ subpopulations (bottom) and qPCR validation (top) of complement

component gene expression in BMDM cultured on GelMA for 24h, compared to BMDM cultured without GelMA (control). For C3, expression in lo versus hi-GelMA in vivo and in vitro. n = 3 (BMDM harvests from individual mice). C3 compared to no GelMA: p = 0.001; C3 lo/hi: p = 0.007 (t-test, two-sided). **e** Fluorescence signal of BMDM seeded on pHrodo-labeled (red) hi- or lo-GelMA for 16h, treated with 10 μg/ml blocking antibodies (CD18, CD11c, CD51, CD11a, and CD11b), 10 μM phagocytosis inhibitor Cytochalasin-D (Cytoch.D), or antibody isotype controls (Isotype). n = 3 (BMDM harvests from individual mice). Left to right: p = 0.01, 0.01, 0.004, < 0.0001, < 0.0001 (one-way ANOVA, Dunnett). **f** Fluorescence signal measurement of BMDM seeded on unlabeled gels stained with lysotracker (green) under conditions similar to (**e**). Left to right: p = 0.05, 0.05, 0.01, 0.01, 0.001 (one-way ANOVA, Dunnett). For microscopy images of (**e**) and (**f**) see Supplementary Fig. 8b. n = 3 (BMDM harvests from individual mice). All data presented as mean ± SD. Source data are provided as a Source Data file.

Supplementary Fig. 12a). By PWD10, wounds treated with hi-GelMA displayed significantly greater fibrotic healing compared to lo-GelMA and Sham, indicated by higher dermal collagen coverage and longer, thicker fibers. Conversely, in lo-GelMA, cellular infiltration was linked with de novo deposition of mature collagen bundles in the gel (Supplementary Fig. 12b), showing its potential for repair and reduced

fibrotic scarring. Of note, normal unwounded skin in P50 mice demonstrated lower metrics of all parameters of fibrosis, including collagen coverage in dermis, and length and width of collagen fibers (Supplementary Fig. 12c).

Fibroblast analysis highlighted contrasting phenotypes and proportions between hi-GelMA and lo-GelMA. hi-GelMA encouraged

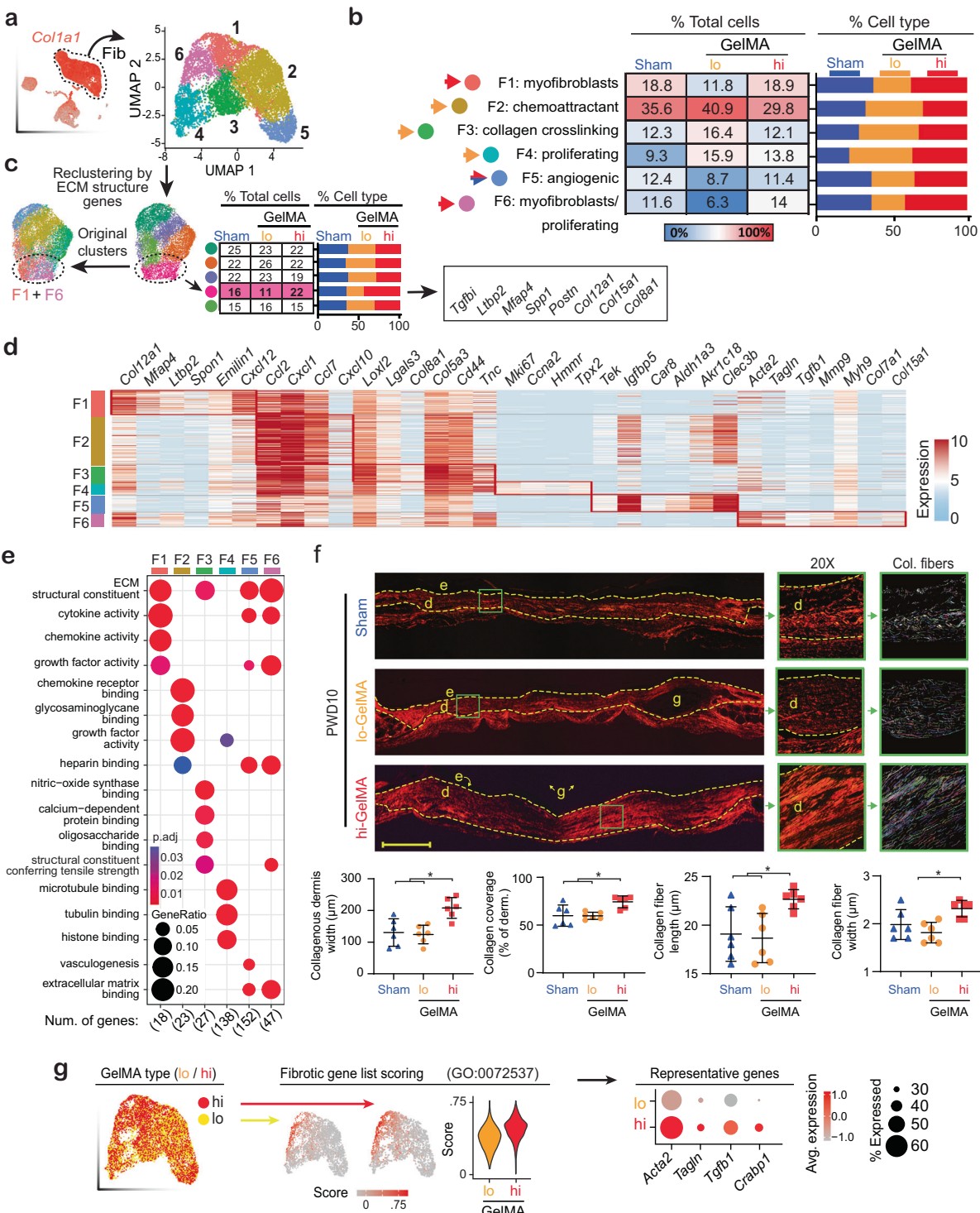

**Fig. 5 | Crosslinking of GelMA hydrogel dressing modulates fibroblast function and collagen deposition in wound healing. a** UMAP representation of fibroblasts from lo- or hi-GelMA and Sham-treated wounds were categorized into six subpopulations presented by UMAP, with each cell color-coded for its associated cell cluster. **b** Table and bar plots showing fibroblast populations of each cluster presented by percentages of total sequenced cells and percentages of specific cell types across treatments. **c** Fibroblasts were re-clustered based on ECM synthesis genes, with two color codes indicating: reclustering based on ECM synthesis genes (right) and original unsupervised clustering (left). A table displaying the percentages of total cells is included, with purple highlight indicating the cluster that aligned with F1 and F6 along with its representative genes. **d** Heatmap showing differentially expressed genes serving as phenotypical markers of each cluster. **e** Dot plot of the enriched GO biological processes of highly expressed genes in

each subpopulation. Dot size corresponds to the proportion of cells within the group expressing each gene, and dot color correspond to GO enrichment p-value. **f** PSR staining of collagen in PWD10 wound sections with green box indicating the magnified region and quantification of dermal collagenous thickness, collagen coverage of dermis and collagen fiber length and width. n = 6 (mice). Derm. Width: hi/S: p = 0.004, hi/lo: p = 0.002, Derm. cover: hi/S: p = 0.01, hi/lo: p = 0.01; Fiber length: hi/S: p = 0.03, hi/lo: p = 0.02, fiber width: hi/lo: p = 0.006 (one-way ANOVA, Tukey's HSD). g: GelMA. Scale bar: 200 µm. **g** UMAP with red (hi-GelMA) and orange (lo-GelMA) mirroring the gene signature score of fibroblast activation (GO:0072537) shown in the feature plot across all fibroblast subpopulations. Violin plots showing gene signature scores in hi- versus lo-GelMA. Dot plot of genes representative of fibroblast activation comparing hi- versus lo-GelMA. All data presented as mean ± SD. Source data are provided as a Source Data file.

pro-fibrotic myofibroblasts, while lo-GelMA promoted chemotactic and repair fibroblasts. To better understand this dichotomy, we analyzed basic fibroblast GO functions using a pro-fibrotic fibroblast activation gene list (GO:0072537). The results showed increased fibroblast activation in hi-GelMA, with higher expression of fibrosis-related genes *Acta2, Tagln, Tgfb1, and Crabp1* (Fig. 5g). In summary, wounds treated with lo-GelMA had more fibroblasts overall, and expression of chemotaxis and proliferation genes. In contrast, wounds treated with hi-GelMA and Sham showed more myofibroblasts and pro-fibrotic gene activation.

## GelMA crosslinking regulates macrophage-fibroblast signaling

As increased GelMA crosslinking led to more inflammatory Mφs and fibrotic fibroblasts, we examined their interplay in hi- versus lo-GelMA-treated wounds. Immunofluorescence staining at PWD5 showed Mφs and fibroblasts in close proximity throughout the wound (Supplementary Fig. 13). CellChat analysis of ligand-receptor pairs revealed heightened outgoing and incoming signals from M4 and reduced incoming signals in M1 for hi- compared to lo-GelMA (Fig. 6a). Information flow analysis, which tracks communication probabilities between cell pairs, showed that hi-GelMA significantly upregulated pro-fibrotic platelet-derived growth factor (PDGF), oncostatin M (OSM), and receptor activator of nuclear factor kappa-B ligand (RANKL) channels (Fig. 6b). Conversely, lo-GelMA prominently featured IL2, GAS, and TENASCIN channels, which are involved in Mφ phenotype regulation.

Considering the Mφ inflammatory signals impact on fibroblast function during tissue repair, we compared receptor and ligand expression between Mφ and fibroblast subpopulations M1, M4, F1, and F6 in hi- versus lo-GelMA (Fig. 6c). In hi-GelMA, M1 and M4 had increased ligand expression of pro-inflammatory (IL1) and pro-fibrotic (TGFβ and OSM) pathways compared to lo-GelMA. M4 also showed elevated pro-fibrotic signaling via PDGF, growth differentiation factor-15 (GDF-15), and insulin-like growth factor-1 (IGF-1). Analyzing specific ligand-receptor gene expression (Fig. 6d), we noted that in the CD137 pathway, M1 and M4 expressed *Tnfsf9*, interacting with F1 and F6 in hi-GelMA due to higher *Tnfrsf9* (CD137) expression. In the IL1 pathway, M1 in hi-GelMA produced more *Il1b*, received by pro-fibrotic F1 and F6 fibroblasts expressing high *Il1r1*, while lo-GelMA fibroblasts expressed higher *Il1r2*, a decoy receptor.

In the TGF-β pathway, M4 in hi-GelMA produced more *Tgfb1* and *Gdf15*, while fibroblast receptor patterns were similar across lo- and hi-GelMA. *Tgfbr2* was expressed higher in lo-GelMA Mφs. In the PDGF pathway, M4 in hi-GelMA produced more *Pdgfb*, which was received by F1 and F6 fibroblasts with high *Pdgfra/b* receptor expression. In the OSM pathway, hi-GelMA M4 produced more *Osm*, which was received by pro-fibrotic F1 and F6 fibroblasts with high *Osmr* expression.

Fibroblasts, especially those in the F1 cluster, acted as ligand expressors signaling to Mφ, with signals including CCL27, CXCL12, IL11, and RANKL (Fig. 6d). Gene expression analysis showed that *Ccl27* and *Cxcl12* in F1 and their receptors *Ccr2* and *Cxcr4* in M1 and M4 were elevated in hi- versus lo-GelMA. F1 and F6 fibroblasts also produced more *Il11* in hi-GelMA, targeting receptors *Il11ra1* and *Il6st* on M1 and M4. In the RANKL pathway, F1 fibroblasts expressed *Tnfsf11*, which targeted *Tnfrsf11a* receptors on M4, both of which were higher in hi-versus lo-GelMA. The FBGC marker *Dcstamp* and Mφ fusion gene scores (GO:0034241) confirmed that M4 responded to RANKL, showing an FBGC phenotype (Fig. 6e). RNAscope targeting *Tnfsf11* (RANKL) and myofibroblast marker *Tagln* on PWD5 slices (Fig. 6f) further validated the *Tnfsf11* source. In hi-GelMA treated wounds, *Tnfsf11* co-localized with *Tagln* significantly more than in lo-GelMA, and aggregated at the GelMA-dermis interface. These data suggest that hi-GelMA scaffolds enhance myofibroblast RANKL signaling, promoting Mφ fusion and FBGC formation.

In lo-GelMA, the IL2, GAS, and TENASCIN channels showed distinct expression patterns. In the IL2 channel, thymic stromal lymphopoietin (*Tslp*), which promotes DC-T cell interactions and type 2 immune polarization, was higher in F1, F6, and M4 of lo-GelMA, while its receptor *Il7r* was evenly expressed across Mφs. *Gas6*, a ligand in the GAS channel, was more prevalent in lo-GelMA fibroblasts, and its receptor *Axl* was elevated in both Mφs and fibroblasts. The *Axl/Gas6* system has anti-inflammatory effects[44] and aids Mφs in clearing debris[45]. In the TENASCIN channel, hi-GelMA Mφs expressed more *Tnc*, which is associated with inflammation[46,47], while lo-GelMA Mφs exhibited higher *Tnxb*, involved in ECM maturation[48]. Altogether, CellChat analysis showed higher pro-inflammatory and pro-fibrotic signals from Mφs to fibroblasts in hi compared to lo-GelMA treatment. In turn, fibroblasts reciprocated with signals promoting FBGC formation in Mφs. Conversely, lo-GelMA Mφs and fibroblasts indicate a shift toward the resolution of inflammation phenotype.

## Discussion

In this study, we investigated the impact of varying GelMA crosslinking on wound healing, uncovering several interaction mechanisms (Fig. 7). Compared to untreated wounds, GelMA dressings led to smaller scars, with softer, faster-degrading lo-GelMA associated with smaller scars, a pro-healing cellular profile, and reduced dermal collagen thickening compared to stiffer, slower-degrading hi-GelMA (Figs. 1, 5). These findings agree with previous studies examining biomaterial stiffness in wound healing[11,12]. Stiffer, highly crosslinked materials such as hi-GelMA provoke stronger FBR, increasing inflammation and fibrosis due to unfavorable cell-material interactions compared to degradable materials. While we used Tegaderm for wound coverage, incorporating strict mechanical controls like wound splinting in future studies will allow for a more accurate evaluation of biomaterial crosslinking effects on healing dynamics. We also observed distinct host responses to lo- versus hi-GelMA, with the former integrated better into the healing skin, degraded faster without triggering FBR, and improved re-epithelialization, dermal proliferation, and reduced scarring (Fig. 1c). Increased porosity in lo-GelMA may have facilitated cellular infiltration, as reports suggest an inverse correlation between GelMA crosslinking and porosity[49]. In contrast, hi-GelMA exhibited early signs of FBR, hindered re-epithelialization, reduced cell infiltration, and was extruded from the wound bed. Contrasting outcomes between lo-GelMA and hi-GelMA highlight the importance of biomaterial properties in wound healing[3,50], and their modulation of immune and stromal cells[51].

The differing reactions to lo- and hi-GelMA prompted us to utilize scRNA-seq to investigate the cellular and molecular mechanisms underlying these outcomes. Analysis of wound cell infiltrate confirmed heightened inflammation and distinct immune cell recruitment and polarization in hi-GelMA versus lo-GelMA (Fig. 1; Supplementary Fig. 2–5). hi-GelMA contained numerous neutrophils, fewer fibroblasts, and increased inflammatory Mφs and mast cells, compared to lo-GelMA and Sham. Conversely, lo-GelMA-treated wounds were enriched with fibroblasts, suggesting more advanced healing effect. These findings align with recent studies that underscore scRNA-seq's crucial role in understanding biomaterial immunomodulation[52], including how different electrospun membrane structures guide cell behavior and influence immune responses[17] and reveal distinct tissue-resident Mφ subpopulations as primary agents of biomaterial degradation[18]. Moreover, the latter study found that scaffold architecture affects mechanotransduction and degradation by Mφs via integrin-dependent mechanisms, consistent with our identification of differential immune cell recruitment and polarization, particularly Mφs, in response to hi- and lo-GelMA.

Mφs are crucial for wound healing but can also contribute to detrimental inflammation or fibrosis, often linked to chronic inflammation, type 1 immune polarization, and FBR to biomaterials[51].

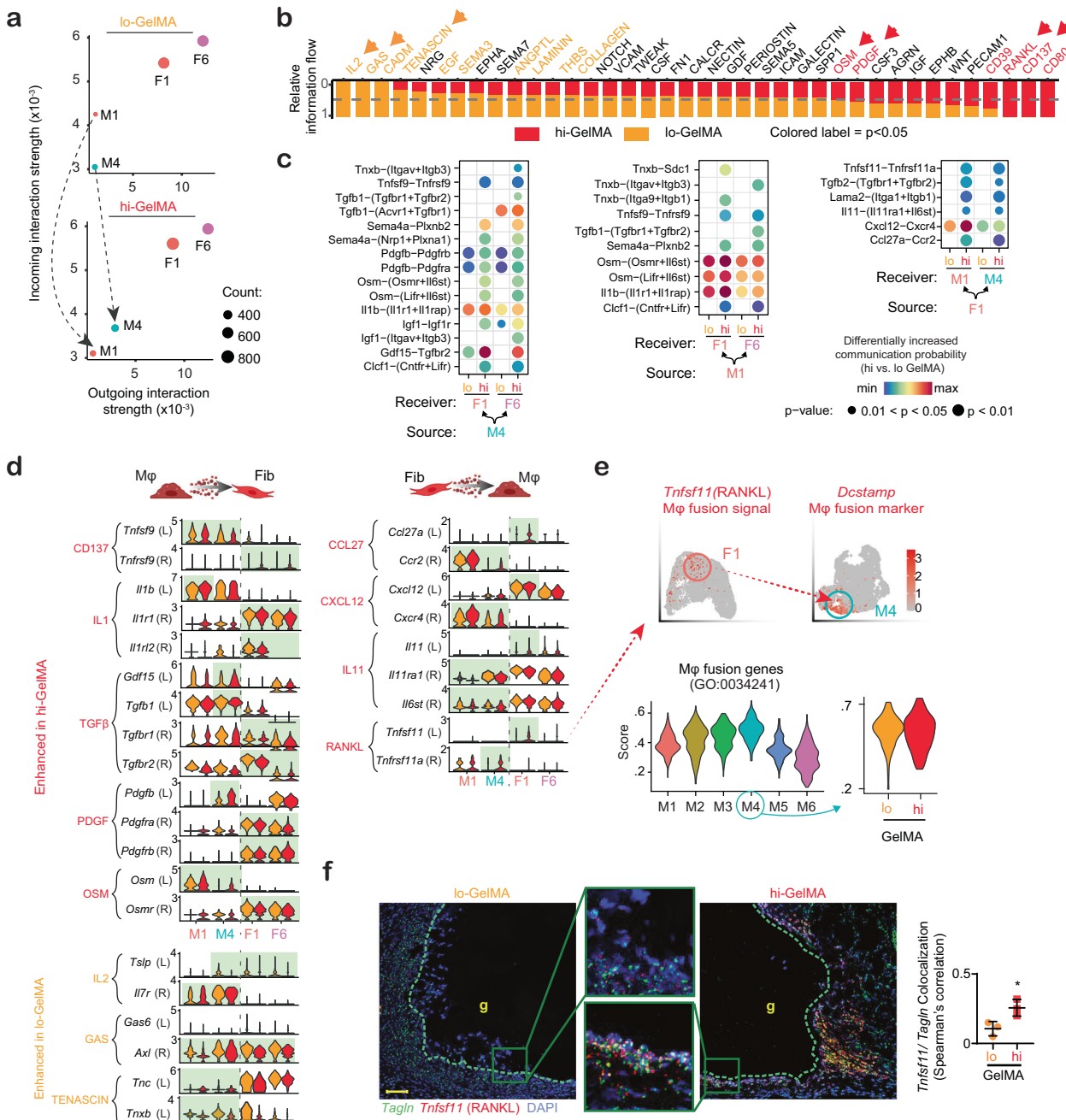

**Fig. 6 | Altered cell-cell signaling interactions between Mφs and fibroblasts in hi versus lo-GelMA treated wounds. a** Two-dimensional visualization representing the overall strength of incoming/outgoing communication in pro-inflammatory M1 and M4 and pro-fibrotic F1 and F6 subpopulations based on receptor and ligand expression, with dot size proportional to the number of cells in each cluster. **b** Comparison of the overall information flow, highlighting significant signaling differences with colored labels. **c** Dot plot illustrating the upregulated signaling ligand-receptor pairs between M1 or M4 and F1 and F6 in hi- versus lo-GelMA, analyzed with permutation tests and adjustments for multiple comparisons to control false discovery rates. **d** Violin plots displaying the expression levels of individual ligand and receptor genes from the signaling channels identified with

differential signaling in M1, M4, F1, and F6, comparing hi- (red) versus lo- (yellow) GelMA. **e** Analysis of the *Tnfsf1*/RANKL pathway and Mφ fusion, presented through feature plots showing F1 as the source of *Tnfsf1* ligand in F1 and the Mφ fusion marker *Dcstamp* in M4, and by gene scoring of the gene list associated with positive regulation of Mφ fusion (GO:0034241). **f** Representative RNAScope images of PWD5 wounds treated with lo versus hi-GelMA stained with probes against *Tnfsf1*/RANKL and *Tagln* (activated fibroblast marker), along with quantification. Green box indicates region magnified. Green dashed line: GelMA-dermis border. g: GelMA. Scale bar: 50 μm. *n* = 3 (mice). *p* = 0.03 (*t* test, two-sided). All data presented as mean ± SD. Source data are provided as a Source Data file.

Preventing FBR is essential for successful tissue regeneration and reduced scarring. Our study shows that high crosslink density enriches pro-inflammatory Mφ subpopulations with increased inflammatory activation (Fig. 2). We found a correlation between GelMA crosslink density, biocompatibility, and cellular response, contrasting with the favorable immune modulation observed with lo-GelMA. Similarly, in

vitro tests showed that hi-GelMA amplified BMDM response to LPS, increasing pro-inflammatory cytokine expression, iNOS levels, and glycolysis. Studies found that DCs on stiffer substrates also had higher inflammatory activation and glycolysis[53]. Our research identified a specific Mφ subpopulation located adjacent to the hydrogels, generating a highly oxidative environment, which was enhanced with hi-

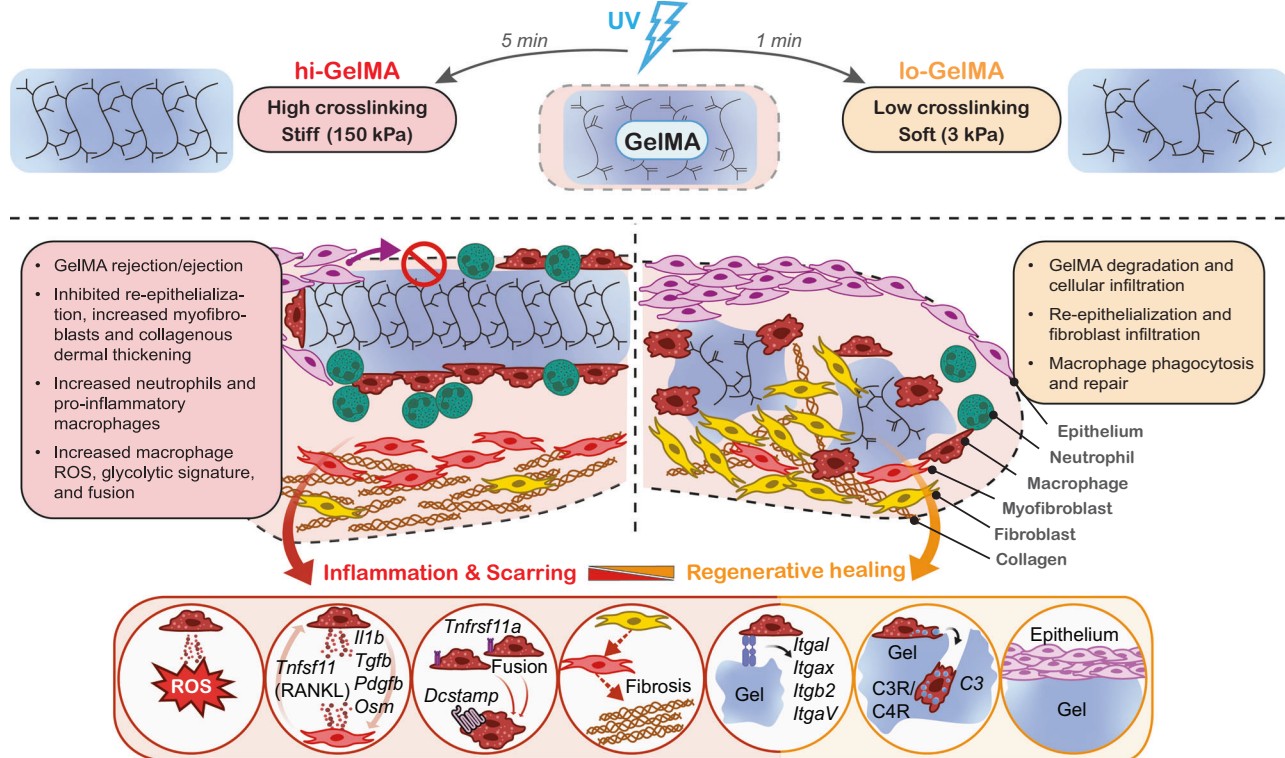

**Fig. 7 | Schematic illustration of the effects of GelMA crosslinking on wound healing.** Low crosslinked (lo-GelMA) hydrogels, treated with 1 minute of UV exposure, are soft (3 kPa) and porous, promoting cell infiltration and integration into wound tissue. Wound repair with lo-GelMA is characterized by Mɸ phagocytosis and pro-healing activities, fibroblast chemotaxis and proliferation, and collagen maturation in the hydrogel, leading to reduced scarring. High crosslinked (hi-GelMA) hydrogels, treated with 5 minutes of UV exposure, are stiff (150 kPa) and nonporous, impeding cell infiltration and subsequent tissue integration. Wounds treated with hi-GelMA display enhanced inflammation and fibrosis, associated with Mɸ pro-inflammation activities and fibroblast pro-fibrotic activation. This response includes Mɸ expressing pro-inflammatory and pro-fibrotic genes *Il1b*, *TGFb*, *PDGFB*, *OSM*, and *Tnfrsf11a* (receptor for the RANKL signal), with pro-fibrotic fibroblasts expressing its ligand *Tnfsf11* (RANKL). Moreover, hi-GelMA promotes Mɸ fusion into FBGC at the tissue-biomaterial interface, leading to elevated levels of ROS and MMPs. Mɸ in contact with the hydrogels exhibit expression of integrins *Itgal*, *Itgax*, *Itgb2*, and *ItgaV* facilitating tissue integration. Figure 7 created with BioRender.com released under a Creative Commons Attribution-NonCommercial-NoDerivs 4.0 International license (https://creativecommons.org/licenses/by-nc-nd/4.0/deed.en).

GelMA (Fig. 3). Proper Mɸ activation is crucial for resolving inflammation and promoting healing[54]. The oxidative Mɸ subpopulations promoted by hi-GelMA likely prolong inflammation, increasing tissue damage and affecting repair. An overly oxidative environment prolongs inflammation, causing tissue damage[55], delayed repair, and increased collagen deposition[56].

We show that GelMA crosslinking affects Mɸ uptake through the C3-CR3/CR4 complement system axis. While lo-GelMA enhances phagocytosis, facilitating the effective clearing of the biomaterial dressing, hi-GelMA promotes oxidative catabolic activities and slower phagocytosis. Storage modulus measurements and imaging of GelMA and cell morphology show that lo-GelMA is softer and more porous, with enhanced phagocytic function associated with a Mɸ dendritic morphology. Conversely, hi-GelMA, characterized by higher stiffness and reduced porosity, leads to a spread Mɸ morphology, indicative of adhesive but reduced phagocytic behavior (Fig. 4; Supplementary Fig. 8, 9). Abundant phagocytosis of lo-GelMA clears the material efficiently and encourages further engagement with additional gel material via the C3-CR3/CR4 axis, creating a self-sustaining phagocytosis cycle, especially in vivo. Our prior work highlighted the complexity of receptor-mediated biomaterial uptake, since knockdown of LAIR-1, a key collagen receptor, led to increased activity in other receptors including CD18 integrin and SRA1[57]. This underscores potential redundancy within these systems, making isolation of specific receptor activity challenging. Similarly, the denatured and heterogeneous composition of GelMA presents a challenge in deciphering the specific contributions of ligand epitopes. The specific interaction

sites of integrins or other receptors and GelMA remain unexplored and warrant further studies to better understand the biomaterial-cell interface, morphological changes in Mɸs and their roles in foreign substance removal.

Our study shows that Mɸs can drive tissue fibrosis through persistent inflammation, type 1 over type 2 immune polarization, and excessive reparative growth factor production, consistent with other findings[51,58,59]. We expand on this by demonstrating that increased wound dressing crosslinking enhances inflammation, leading to an abundance of pro-fibrotic fibroblasts (Fig. 5). Conversely, lo-GelMA reduced pro-inflammatory immune cells and increased chemotactic and proliferative fibroblasts, indicating progression beyond inflammation. Although hi-GelMA and Sham wounds contained fewer fibroblasts overall, they had more pro-fibrotic activation and developed a thicker collagenous dermis, while lo-GelMA-treated wounds showed greater cellular infiltration, and formed new collagen bundles within the gel matrix (Fig. 5f). This suggests that the inflammatory and fibrotic response to hi-GelMA versus reduced inflammation and scarring response to lo-GelMA likely involves Mɸ and fibroblast activities, as well as their interactions.

Analysis of cell-cell communication networks highlighted IL1, CD137, TGFβ, PDGF, and OSM as signaling channels between Mɸ and fibroblasts that pushed pro-fibrotic fibroblast activation in the hi-GelMA treatment condition (Fig. 6). Recent findings indicate that the OSM pathway is crucial in Mɸ-fibroblast crosstalk during early wound healing, significantly upregulating fibroblast activation genes[60]. Enhanced CD137 signaling in hi-GelMA was due to higher expression of its

receptor *Tnfrsf9* in fibroblasts, which is associated with inflammatory activation[61,62]. Conversely, fibroblasts in lo-GelMA expressed higher *Il1r2*, a decoy receptor that inhibits *Il1r1* inflammatory signals[63], and lo-GelMA Mφs expressed more receptor *Tgfbr2*, suggesting a shift toward inflammation resolution. In hi-GelMA, F1 fibroblasts transmitted CCL27, CXCL12, IL11, and RANKL inflammatory signals to Mφs, stimulating Mφ fusion and forming FBGC. M4 Mφs in hi-GelMA expressed higher *Tnfrsf11a*, along with the FBGC marker *Dcstamp*, indicating that these Mφs are fusing as part of an enhanced FBR. This suggests that increased inflammation in hi-GelMA Mφs is linked to a heightened FBR, where Mφs fuse to form FBGC, producing elevated ROS levels[51]. A recent study using scRNA-seq to analyze FBGC responses to stiff implants aligns with our findings, showing that FBGC adjacent to hi-GelMA express genes related to glycolysis, *Mmp12*, and *Dcstamp*[64]. The role of RANKL signaling in Mφ fusion and the amplified FBR and fibrotic response to hi-GelMA underscores the need for further exploration of stiffness or crosslinking-dependent FBGC formation and RANKL signaling.

Signals associated with reduced scarring were enhanced in lo-GelMA versus hi-GelMA. *Tslp*, which promotes DC and T cell immunity through the *Il7r* receptor[65], is also involved in type 2 responses via innate lymphocytes and T cells, in coordination with IL-25 and IL-33[66]. Our study found higher *Tslp* expression in fibroblasts and *Il7r* expression in macrophage T cells, and DCs in lo-GelMA compared to hi-GelMA, suggesting adaptive immune involvement. Studies with other regenerative biomaterials have identified both IL-33[67] and type 2 immune polarization[68] as potential drivers of immune programs. Furthermore, *Gas6* expression was higher in fibroblasts, and *Axl* expression was increased in both Mφs and fibroblasts in lo-GelMA. *Axl* encodes a TAM receptor tyrosine kinase that transmits signals by binding to *Gas6* and executes an anti-inflammatory effect[44]. Heightened Axl/Gas6 signaling, which supports Mφ debris clearance and inflammation resolution[45], is consistent with the greater clearance and pro-resolution effects of lo-GelMA. Finally, hi-GelMA promoted *Tnc*, encoding Tenascin C, associated with inflammation and TLR4 signaling[46,47], while lo-GelMA promoted *Tnxb*, encoding Tenascin XB, which supports collagen ECM maturation[48]. These opposing patterns are consistent with differential outcomes based on GelMA crosslinking and recent work identifying TNC as influential in Mφ-fibroblast crosstalk during wound healing[60].

Our findings show that hydrogel crosslinking shapes wound healing by influencing Mφ and fibroblast interactions, revealing distinct inflammatory profiles and complex bidirectional signaling. Further research into GelMA-tissue dynamics will inform the development of other regenerative hydrogels. Understanding how surface structures and crosslinking impact cell interactions and foreign body recognition will aid in designing biomaterials with optimal biocompatibility for improved wound healing.

## Methods

### Ethical statement
All experiments involving animal use, husbandry, and wounding were reviewed and approved by the relevant regulation authority, the IACUC of the University of California, Irvine. The experiments were performed in accordance with relevant guidelines and regulations.

### Gel fabrication
Lyophilized gelatin methacrylate (GelMA) (Advanced Biomatrix) was reconstituted at 20% w/v with >60 °C PBS; 10% Irgacure 2959 dissolved in methanol was added to GelMA to a final concentration of 0.01%. This material was kept at 37 °C until cast onto sterile coverslips or in situ on murine dorsal 5mm full-thickness skin wounds. GelMA stiffness was characterized by a parallel plate rheometer on a DHR3 instrument (TA instruments). Briefly, 500 µl of GelMA solution was pipetted onto the stage, and the 40mm plate was then lowered to 300mgap before 365nm UV crosslinking from below the stage. An amplitude sweep was conducted from strain of 0.01–10%, to measure storage modulus of the hydrogel. 4W 365nm UV light exposure for 1 min yielded soft 3 kPa gels (lo-GelMA), and 5 min yielded stiff 150 kPa gels (hi-GelMA) (Supplementary Fig. 9a). For fluorescent imaging, gels were incubated with fluorescein-NHS-ester (Thermo Fisher) O.N at 4°C, followed by wash and imaging using scanning laser confocal microscope.

### Wounding studies
Full-thickness skin wounding was conducted on p50 C57BL/6J female mice (Jackson Laboratory). Mice were anesthetized using isoflurane and shaved; p50 mice were chosen to minimize the impact of hair follicles during wound healing. Dorsal skin was cleansed using 70% ethanol, and a single full-thickness wound was made with 5mm biopsy punches at the dorsal midline, immediately below the scapulae. This location was chosen to minimize disruption to the wound during healing. Wounds were treated with 20ul 20% GelMA, and UV cross-linked for either 1 (soft, 3 kPa) or 5 (stiff, 150 kPa) minutes. Wounds were then dressed with Tegaderm (3M), followed by two ¾"x2" adhesive flexible bandages (Band-Aid). Of note, wounds were not splinted by external rings. Mice were housed individually after wounding. Mice were monitored daily for signs of infection/healing. At 3-, 5-, 10-, or 30-days post-wounding, mice were sacrificed, and dressings carefully removed. Wounded skin was excised with a ≥5mm margin and mounted in OCT for cryo-sectioning or fixed in 10% formalin overnight at 4°C for formalin fixed paraffin embedded sectioning or whole-mount imaging with a dissection microscope.

### Scar size analysis
PWD30 samples were carefully shaved, and the whole back skin was dissected out, fixed in 10% formalin overnight at 4°C, and imaged with a dissection microscope. Scarring was quantified as the area of the scar, defined by the wound edge, which was marked where the hair shaft meets the skin surface.

### Histology
Formalin-fixed paraffin embedded tissues were cut into 5 µm sections on histological slides. Then, stained with hematoxylin and eosin (H&E) and Picrosirius Red (PSR). Imaging was performed with Olympus FV3000. PSR images were taken under polarized light settings. Measurements were conducted using Fiji-ImageJ software. For quantifying re-epithelialization, H&E data were utilized; this involved measuring the distance covered by the migrating epithelium from the edges of the original full excisional wound towards the center in each sample. Additionally, the quantification of GelMA area was performed using H&E staining, involving manual marking and measurement of the GelMA area. PSR staining was used for the quantification of the collagenous dermis width, collagen coverage as a percentage of the dermis, and collagen fiber length and width, by averaging three regions of interest per sample. Dermis width was measured from the epidermal-dermal junction to the dermal-hypodermal junction. Measurements of collagen fiber length and width were conducted using the CT-FIRE plugin[69].

### Bone-marrow-derived macrophage (BMDM) culture
Femur and tibia bones were harvested from C57BL/6J female mice (Jackson Laboratory), and bone marrow was flushed with PBS, centrifuged, and resuspended in ammonium-chloride-potassium (ACK) lysis buffer (Thermo Fisher) to lyse red blood cells for 1 min. Then, blocked with Dulbecco's Modified Eagle's medium (DMEM) supplemented with 10% heat-inactivated fetal bovine serum (FBS) and centrifuged. Pellet was resuspended and cultured in high-glucose DMEM supplemented with 10% heat-inactivated FBS, 2 mM ʟ-glutamine, 100 U/ml penicillin, 100 µg/ml streptomycin (Thermo Fisher), and 10% conditioned media from CMG 14–12 cell expressing recombinant mouse Mφ colony stimulating factor (M-CSF). The generated

conditioned media stock was diluted with complete DMEM at a ratio of 1:10, mixing 50 mL of conditioned media with 500 mL of DMEM. This typically results in the concentration of 35,000 U/ml M-CSF[70]. Cells were cultured for 7 days to induce differentiation to BMDM.

## ELISA

BMDM were stimulated with 10ng/ml ultrapure LPS (Invivogen). Supernatants were collected 6 h post-stimulation for assessment of cytokine secretion by standard enzyme-linked immunosorbent assay (ELISA) following the manufacturer's protocol (Biolegend).

## Immunofluorescence staining

For cell cultures, BMDM were fixed in 4% PFA for 10 min at room temperature (RT). The cells were washed three times with 1XPBS for 10 min each. Then the cells were permeabilized using 0.3% Triton X-100 in PBS for 10 min at RT. After washing the cells as mentioned earlier, they were incubated with primary antibody overnight at 4°C with shaking, Arginase (Abcam 60176 1:50) and iNOS (Abcam 15323 1:100). The cells were washed with 2% BSA-1XPBS and incubated with secondary antibody Thermo A21209, A21244, A21206) at 1:1000 dilution, at RT for 1 h. The nuclei and actin were stained using 1:2000 Hoechst and 1:200 Alexa fluor 488-Phalloidin (Invitrogen), respectively, diluted in 2% BSA-1XPBS for 30 min at RT. Finally, the cells were washed with 1XPBS for three times, 10 min each at RT and mounted on the glass slide using Fluoromount G (Southern Biotech) and imaged using the Olympus FV3000 laser scanning microscope. The quantification of F4/80 and iNOS signals was done by measuring the percent of F4/80 / iNOS positive cells throughout the total wound area.

For tissue sections, frozen tissue sections were thawed to room temperature and fixed in 4% PFA (Fisher) for 15 min, then washed in 4 changes of PBS (VWR). Tissues were permeabilized with 0.1% Triton-x-100 (Sigma) and then washed three times with PBS 0.1% Tween-20 (Sigma), five minutes each, before blocking in 1% BSA (MP Biomedical 0219989880) + 0.1% Tween for 2 hours. Sections immunostained with F4/80 (Thermo MF48000 BM8 1:200), Arginase (Abcam 60176 1:50), iNOS (Abcam 15323 1:100), PDGFR-α (Abcam AF1062 1:200) overnight 4°C. Slides were then washed three times with PBS 0.1% Tween-20, 10 minutes each, and then stained for 1 hour with fluorescent conjugated secondary antibodies (Thermo A21209, A21244, A21206) and Hoechst 33342 at 1:1000 dilution. After again washing three times with PBS 0.1% Tween-20, five minutes each, slides were mounted with Fluoromount (Southern Biotech) and imaged using the Olympus FV3000 laser scanning microscope.

## qPCR

mRNA expression was quantified using the following primers:

| Gene | Forward | Reverse |
|---|---|---|
| Itgal | TGCAGCCTATCCTGAGACCT | AGTGTCCACTCCACAGCAAG |
| Itgam | TCCGGTAGCATCAACAACAT | GGTGAAGTGAATCCGGAACT |
| Itgax | CTGGATAGCCTTTCTTCTGCTG | GCACACTGTGTCCGAACTCA |
| Itgad | GGAACCGAATCAAGGTCAAGTA | ATCCATTGAGAGAGCTGAGCTG |
| Itga2 | GCGGCAGAGATCGATACACA | CTTCTGCTTTCTCCGTGGGT |
| Itga4 | AGGCAGAGTCTCCGTCAAGA | GGCCTCTACATGAATGGGGG |
| Itga5 | CAAGGTGACAGGACTCAGCA | GGTCTCTGGATCCAACTCCA |
| Itga6 | ATCCTCCTGGCTGTTCTTGC | CAGCCTTGTGATAGGTGGCA |
| Itgb2 | CAGATTCTCGGAGTGGAGGC | ACTTGGTGCATTCCTGGGAC |
| Itgb1 | AGGTCGATCCTGTGACCCAT | ATGTCGGGACCAGTAGGACA |
| Itgav | CCGTGGACTTCTTCGAGCC | CTGTTGAATCAAACTCAATGGGC |
| Itgb3 | GTGGCCGGGACAACTCTG | GGACTCACAGCCAGACACTG |
| Itgb5 | TCCAGGGCCCGTTATGAAAT | CACGCCAGAGTCTTCATCCT |

## Single-cell RNA-sequencing

PWD5 wound tissue was dissected from the surrounding edge and healthy tissue, minced with scissors, and then dissociated with 10 ml solution of 2.7 mg/mL collagenase P (Sigma), 1 mM pyruvate, 10 mM HEPES, in RPMI basal media at 37°C for 2 hours at 37°C, triturating every 15 minutes. Digestion was halted with the addition of 2ml 2% FBS and live cells were isolated using Miltenyi dead cell separation kit and MS columns, per manufacturer's instructions. Cells were finally suspended in 1ml 0.04% BSA and kept on ice until processed for library preparation. RNA library was prepared using a 10X Chromium V3.1 kit and sequenced with NovaSeq on an S4 flow cell (UCI Genomic High Throughput Facility).

## Sequencing data analysis

10X Chromium sequencing FASTQ output files were aligned using Cell Ranger. Data was filtered for quality control, removing cells with greater than 5% mitochondrial DNA content, more than 7500 or fewer than 200 genes expressed. After quality control, we retained ~17,000 cells for downstream bioinformatic analyses. Clustering was performed using the Seurat R package[71]. Samples were integrated with SelectIntegrationFeatures function with 3000 genes and downstream integration functions of PrepSCTIntegration, FindIntegrationAnchors and IntegrateData using SCTransform. Prior to clustering, we performed dimensionality reduction using Principal Component Analysis (PCA). To identify significant principal components (PCs), we used the JackStraw function, selecting the top 50 PCs. A shared nearest-neighbor graph was constructed using the PCA embedding. Clusters were identified using the Louvain modularity-based community detection algorithm underlying the FindClusters function, setting resolution = 1. Marker genes were determined using FindConservedMarkers or FindMarkers with test.use = "wilcox"; $p$-value < 0.01 and log(fold-change) > 0.25. For visualization, RunUMAP was used with reduction = "pca", dims = 1:50. Mφ clusters were subset and unsupervised clustering was used to identify Mφ subpopulations. This process was repeated for fibroblasts. Mφ and Fibroblast subsets were analyzed with FindMarkers, to characterize cluster identities and DGE between treatment conditions. Differentially expressed genes were analyzed by GO analysis in clusterProfiler[72] to identify putative signaling pathways. Annotations were generated using AnnotationHub and EnsDb.Mmusculus.v79. Single-cell gene signature scoring was generated with Ucell[31]. CellChat cell-cell interaction network analysis[38] was used to characterize Mφ-fibroblast interactions. The pathways identified were then assessed for their contribution to receptor/ligand interactions.

## mRNA in situ hybridization

Fresh frozen tissues were cut into 5 µm sections on histological slides. Slides were fixed and manual RNAScope was performed using multiplex fluorescent reagent kit v2 and company protocols (Advanced Cell Diagnostics). RNAScope probes: negative control probe (320871), positive control probe (320881), Tnfsf11 (410921), Tagln (480331-C2), Fcgr1 (487701-C3) and Cybb (403381-C2). Detection of probes was done using RNAScope recommended reagents: OPAL 520, OPAL 570, OPAL 690 (Akoya Biosciences). Colocalization analysis done with Fiji-ImageJ software using Coloc 2 plugin to quantify colocalization between two color signals, employing Spearman's rank correlation coefficient. This non-parametric approach assesses the ranked relationship between signal intensities, suitable for our non-linear data. The analysis was complemented by the visual inspection of the merged images to confirm colocalization areas.

## In vitro ROS and mitochondria staining

GelMA substrates (20% gel, 0.1% Irgacure 2959) were prepared on a glass slide and covered with a coverslip, polymerized with UV for 1m

(lo-GelMA) and 5m (hi-GelMA). BMDM were seeded at a density of 0.25E6 cells per well in 24-well plates, on GelMA substrate for 24 h before stimulation with 1 μg/ml LPS (Sigma) for 16 h. For ROS staining, 5 μM CellROX-green dye (Invitrogen) was added in complete medium for 30 min at 37°C, then cells were counterstained with Hoechst 33342 and imaged alive. For mitochondria staining, MitoTracker Red CMXRos (Invitrogen) was added for 30 min, cells were fixed and mounted. Imaging was performed with Olympus FV3000 laser scanning microscope. Mitochondria morphological analysis was done with Fiji-ImageJ using particle analysis for individual mitochondrial particles and Mitochondrial Network Analysis (MiNA) toolset[73] for branched mitochondrial network particles.

### In vivo ROS activity
Mice were wounded with a 5mm punch through a folded back skin creating a pair of wounds. Each wound was cast with either lo or hi-GelMA. Luminol sodium salt (CAS: 20666-12-0) dissolved in sterile saline (0.9% NaCl) at 10 mg/ml and injected I.P to a final dosage of 100 mg/kg. Mice were injected a day after wounding and imaged immediately with In vivo Imaging System (IVIS).

### Phasor autofluorescence NADH lifetime imaging
Fluorescence Lifetime Images are acquired with a two-photon microscope coupled with ISS FLIMBOX Spartan 3, Ti:sapphire laser (Spectra-Physics Mai Tai) with 80 MHz repetition rate is used to excite the sample. The laser is coupled with a Zeiss LSM 880 microscope. Excitation is 740 nm (two-photon excitation), objective lens with 690 nm band pass filter, immersion objective is Zeiss C-Apochromat 40× water NA 1.2 and laser power 5mW. For image acquisition the following settings are used: image size of 256 × 256 pixels, pixel size 230 nm, scan speed of 16.38 μs/pixel acquiring and averaging 30 frames (about 2s each frame). Emission signal directed to the external hybrid detector (H7422P-40, Hamamatsu Photonics, Hamamatsu, Japan) coupled to FLIM-box (ISS, Champaign, IL, USA). NADH fluorescence signal is acquired through a 460/80 nm band 247 pass filter. The images are acquired and processed with SimFCS software developed at LFD. The phasor transformation is reported in the supporting information. FLIM calibration is conducted using a 100 μM solution of Coumarin 6 in ethanol (single exponential lifetime 2.5 ns). NADH free/bound analysis is performed using SimFCS software, as reported previously. Briefly, if a pixel contains two species with two different fluorescence lifetimes, the phasor will appear along a straight line joining the phasor coordinates of the two lifetimes. The position of the phasor along the line will depend on the relative brightness contribution of each species. Using the model derived from the linear unmixing, the NADH free/bound ratio is measured.

### GelMA uptake
GelMA substrates were prepared as outlined in the gel fabrication section, supplemented with 2 μM pHrodo (Thermo Fisher). We cross-linked 50 μl gels between a slide and a 15 mm coverslip to create flat discs, placed them in a 24-well plate, and seeded 0.3E6 BMDM for 16 h. Post-incubation, BMDM were stained with CellTracker Green CMFDA (Thermo Fisher) and imaged using an Olympus FV3000 microscope. For blocking experiments, hydrogels were fabricated similarly and treated with 10 μg/ml blocking antibodies: CD18 (clone M18/2, Abcam 119830), CD11c (Clone N418, Abcam 33483), CD11b (Clone M1/70.15, Biolegend 101202), CD51 (Clone RMV-7, Thermo Fisher 14051282), and CD11a (Clone M17/4, Thermo Fisher 14011182), plus isotype controls (Biolegend 400622, 400516, 400123) and 10 μM Cytochalasin D (Abcam 143484). Phagocytosis was evaluated by staining hydrogels with pHrodo or using unstained gels, with BMDM marked with Lyso-tracker green (Invitrogen). Stained gels were uniformly exposed to UV, and phagocytosis measurements were taken at 16 h with a Varioskan

LUX plate reader (Thermo Scientific), using cell-free hydrogels as blank controls. Adhesion and morphology assessments involved Hoechst and phalloidin green staining, respectively, imaged by fluorescence microscopy. Morphology analyzed using Fiji-ImageJ software MorphoLibJ[74].

### Statistics
Data are presented as individual points + mean ± standard deviation (SD). The statistical significance of differences between indicated samples was determined using the following methods. For normally distributed data, comparisons between two groups were performed using either the paired or unpaired Student's $t$ test, as appropriate. For comparisons involving multiple groups, one-way ANOVA followed by Tukey's post-hoc test was employed. For correlation analyses of non-normally distributed data, the Spearman method was utilized. All tests were conducted using GraphPad Prism 8 software. $p$ values, sample sizes ($n$), and whether tests were two-sided are indicated in the figure legends. In animal experiments, the choice of five mice per treatment group was made to balance statistical power with ethical considerations in animal usage. We increased this number to 6 or 9 in cases where variability in wound data warranted a larger sample size to ensure statistical reliability. These adjustments in sample sizes were made to enhance data robustness and reliability while adhering to ethical guidelines for research.

### Reporting summary
Further information on research design is available in the Nature Portfolio Reporting Summary linked to this article.

## Data availability
All data supporting the findings of this study are available within the article and its supplementary files. Any additional requests for information can be directed to, and will be fulfilled by, the corresponding authors. Sequence raw data are available in the NCBI Gene Expression Omnibus (GEO) under accession number GSE248524. Source data are provided with this paper.

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

## Acknowledgements

This work was supported by the National Institutes of Health: National Institute of Allergy and Infectious Disease (NIAID) Grant Number 1R01AI151301 (W.F.L.), National Institute of Arthritis and Musculoskeletal and Skin Diseases (NIAMS) Grant Number 1R21AR077288 (W.F.L.), P30AR075047 Skin Biology Resource Center Seed Grant (W.F.L., M.V.P., Q.N.), R01AR079150 and U01AR073159 (Q.N. and M.V.P.), National Science Foundation grant DMS1763272 (Q.N.), a Simons Foundation grant (594598, Q.N.), LEO Foundation grants LF-AW-RAM-19-400008 and LF-OC-20-000611 (M.V.P.), Horizon Europe grant 101137006 (M.V.P.) and W.M. Keck Foundation grant WMKF-5634988 (M.V.P.). We also acknowledge NIH Office of Director (OD) Grant S10OD025064 to E.L.B. for support of a confocal microscope facility, which enabled imaging work. The authors used Biorender (http://biorender.com) in the creation of the figures.

## Author contributions

S.B., R.R.N., C.G.J., F.P., L.D., and R.Q.N. performed experiments and analyzed the results. D.G., A.A.A., M.A.D., Q.N., P.O.S., M.V.P., and W.F.L. assisted in analysis and writing the manuscript. S.B., R.R.N., M.V.P., and W.F.L. planned the experiments, analyzed the data and wrote the manuscript.

## Competing interests

The authors declare no competing interests.
