## [Peer Review File · Nature Communications]

REVIEWER COMMENTS

Reviewer #1 (Remarks to the Author):

The author investigated how the degree of crosslinking in two gelatin methacrylate (GelMA) hydrogels affects wound healing. The hydrogels had different levels of crosslinker, and the author used single-cell RNA sequencing, gene ontology analyses, and cell-cell interaction network analysis to determine the cell infiltrate in the wound site, different subpopulations of macrophages and fibroblasts, and the main processes enriched when comparing low and high degrees of crosslinker. The author hypothesized and demonstrated that stiffer gels with high crosslinkers promote a more pro-inflammatory response and a pro-fibrotic phenotype. While softer hydrogels with fewer crosslinkers create a more reparative environment.

The authors utilized an approach pairing single-cell RNA sequencing, histology analysis, and cell-cell interaction network analysis complementary with in vitro experiments using BMDM to corroborate their functional enrichment results. Though the combination of these approaches is innovative, the rigor of all the approaches were not equal, limiting the conclusions that could be made. In particular, while scRNAseq can be robust with one biological run combining cells from 5-wounds, histological analyses cannot be done from a single wound and the rigor of the in vitro experiments are not clear. Starting with the histological analysis, it appears much lower in rigor than the rest of the results. Overall, multiple images, wounds, animals should be analyzed to assess reproducibility. In particular:

- Why does the number of animals vary for each in vivo experiment? This seems to vary without explanation
- The H&E images provided make it difficult to differentiate between the layers of the skin (Fig 1C).
- The location of the immunofluorescence images is not clear, especially for Fig 2E and Fig 5a. For example, in Fig 5a, PWD5 and PW10 seem to be in high GelMA, but the authors mentioned that high GelMA tends to extrude outside the skin due to foreign response. How many wounds, sections per wound, ROIs per image were analyzed? Was an unbiased quantification approach used?
- The author used Picosirius red Fig 4E to analyze collagen deposition at PWD10 but only quantified the collagenous dermis width between the treatments without quantifying the percentage of collagen or other characteristics of collagen fibers and without comparing with controls (sham and normal skin). This seems like a huge omission given the title of the paper involving fibroblasts.

Turning to in vitro experiments. The authors must expand the statistical analysis for their in vitro experiments to provide more comprehensive results. It is also unclear whether the in vitro results are biological or technical replicas.

Additional comments:

- To ensure there is no batch effect in the experiment, could the authors perform principal component analysis (PCA) or hierarchical clustering on the full set of data, as the results presented in the scRNA-seq were pooled from five wounds?
- Could you provide more information on the parameters that were utilized to determine the highlighted area in Figure 1b? Based on the visual data, it appears that the yellow highlighted area may be subjective as there seems to be hair present in the center of the Sham wound.
- In the experiment, was the full excisional model used and was it splinted? There is no information provided in the methodology section. If it was not splinted, how did the authors prevent local skin contraction to ensure that the wound was healed mainly by granulation and re-epithelialization?
- Specify the concentration of recombinant mouse M-CSF used in the methodology section.
- The methodology section should include details on how to calculate collagen width, iNOS, Arg+1 expression, and ROS activity.
- Is the in vitro data a biological replica or a technical replica (with gels being replicated on the same day)?
- The methodology section states that the wound samples were analyzed at 3, 5, 10 and 30 days post-wounding. However, there is no information provided for day 30 in the manuscript.

Reviewer #2 (Remarks to the Author):

The paper discusses the changes in wound healing and related cellular dynamics in mice due to different cross-linking of GelMA, including the relationship between macrophages and fibroblasts, and the effects of hypoxia. It is interesting to note the many single-cell analyses and discussions based on them. However, in this experiment, the difference between hi-GelMA, which is eliminated from the wound area early after transplantation, and lo-GelMA, which remains intact in the wound area, is thought to have a significant effect on the final scar. It is difficult to explain the difference in scar formation when there is a difference in whether or not the foreign material remains in the body. In the case of hi-GelMA, since the blood vessels have not yet invaded the matrix to be eliminated, there is no re-epithelialization on top of this foreign material, and it is difficult to attribute this to a difference in epithelialization due to the difference in cross-linking. The paper should be written from the viewpoint of the difference in biocompatibility and the mechanism of recognition of foreign substances, and from the viewpoint of the relationship between their surface structures and

molecular interactions between cells to examine why lo-GelMA and hi-GelMA are more or less likely to be eliminated as foreign substances. We believe that the paper should be written in the direction to consider the relationship between the surface structure and molecular interaction between cells. For this reason, I do not consider it suitable for publication in Nature Communications.

Reviewer #3 (Remarks to the Author):

This paper describes cell responses on gelatin methacrylate (GelMA) hydrogels with different crosslinking densities. Authors prepared lightly crosslinked (lo-GelMA) or heavily crosslinked (hi-GelMA) hydrogels and evaluated their responses of macrophage and fibroblast using single-cell RNA-sequencing. Data are interesting and informative, however, there are lack of novelty of materials and methods for analysis of cell response. So, reviewer recommends to submit more specialized journal.

Point by point response to reviewers

Reviewer 1

- Why does the number of animals vary for each in vivo experiment? This seems to vary without explanation.

We thank the reviewer for raising this question. Scar size was found to be highly variable, requiring larger numbers of animals (n=9) for statistical comparison, whereas other analyses were less variable, and thus, fewer animals were analyzed. For immunohistochemistry, we used frozen samples with n=5. For other analyses, we used paraffin-embedded samples with n=6. All details of sample numbers have been added to the figure legends and rationale have been added to the manuscript in the statistical methods section (lines 781-793).

- The H&E images provided make it difficult to differentiate between the layers of the skin (Fig 1C).

We thank the reviewer for this insightful comment. In response, we made several additions. To Fig. 1c (H&E), we added yellow dashed lines to delineate borders between epidermis and dermis, as well as between dermis and hypodermis. We have also included a new Supplementary Fig. 1 that contains additional histological images.

- The location of the immunofluorescence images is not clear, especially for Fig 2E and Fig 5a. For example, in Fig 5a, PWD5 and PW10 seem to be in high GelMA, but the authors mentioned that high GelMA tends to extrude outside the skin due to foreign response. How many wounds, sections per wound, ROIs per image were analyzed? Was an unbiased quantification approach used?

We thank the reviewer for pointing out this oversight. Although we included full aspect of the wound in the supplementary figure, we did not indicate the location of enlarged images shown in Fig. 2e. We have now added squares to Supplementary figure 6 to denote the location of enlarged images in Fig. 2e.

Regarding Fig. 5a, which has been moved to Supplementary Fig. 12 in the revised manuscript, we have shown IHC of macrophages and fibroblasts from representative ROIs of PWD5 sections, highlighting their proximity and density across the wound and near the hydrogel (denoted with a g). Indeed, we observed that hi-GelMA is typically extruded from the wound but not until PWD10. We replaced image in Fig. 1c (PWD10/hi-GelMA) for a better representation of the location of gel remnants as they are ejected from the wound. Notably, at earlier stages, such as PWD3, we detected abundant immune cells surrounding the gel and between gel and Tegaderm dressing above it.

To improve clarity, we made the following modifications to this panel, now Supplementary Fig. 12:

- It now provides a large field of view of the full aspect of wounds.
- Focus is on PWD5, the most relevant stage to our scRNA-seq data.
- It includes a magnified view of ROIs.

- Images with DAPI are provided to clarify the morphology of the wound and gel.
- Images without DAPI are included to clarify co-staining of macrophages and fibroblasts.

We also thank the reviewer for emphasizing the need for clarity in our methodology. We have elaborated on the quantification strategy in greater detail in the Methods section (lines 679-80). Briefly, the study involved examining five wounds per mouse for each condition. For quantification, we analyzed one section per wound, covering the entire wound bed in each image. All immunohistochemistry images were collected in a single session for each time point and were thresholded for consistent contrast. DAPI staining was used to count all cells, and cells positive for other stains were counted separately, enabling us to calculate the percentage of positive cells.

- The author used Picosirius red Fig 4E to analyze collagen deposition at PWD10 but only quantified the collagenous dermis width between the treatments without quantifying the percentage of collagen or other characteristics of collagen fibers and without comparing with controls (sham and normal skin). This seems like a huge omission given the title of the paper involving fibroblasts.

We thank the reviewer for their valuable comment. In response, we have implemented several updates. Firstly, in Fig. 4f, we replaced Picosirius Red staining images with higher-resolution versions and included Sham sample for comparison. Description of the methods used for measurements has been enhanced in the Methods section (lines 647-651). Additionally, Fig. 4f now features magnified view of the collagen bundles. This new imaging allowed measurements of collagen coverage as a percentage of the dermis area, as well as the length and width of the collagen fibers, which is now also included in the same panel. Detailed information about the methodology for these complementary analyses has also been added to the Methods chapter, under the subtitle "Histology" (lines 651-652). Furthermore, in alignment with H&E data, we also included additional Picosirius Red stains in the supplementary figures (Supplementary Fig. 11).

We appreciate the reviewer's suggestion to include normal skin as a control. Our study primarily focuses on comparing healing and treated wounds, with Sham controls providing relevant baseline for these conditions. Including normal skin, while informative, might not directly align with our study's focus on the wound healing process and treatment efficacy, given the significant differences in collagen architecture in unwounded skin. However, to address this point, we have added data from four normal skin samples (from four mice) to our analysis, including quantifications of collagen coverage in the dermis, and collagen fiber length and width (Supplementary Fig. 11c).

- Turning to in vitro experiments. The authors must expand the statistical analysis for their in vitro experiments to provide more comprehensive results. It is also unclear whether the in vitro results are biological or technical replicas.

We thank the reviewer for highlighting the need for clarification on our data collection methods and statistical analyses. For the in vitro experiments shown in Figs. 2g, 2h, 3g, and the newly added Fig. 3k (phagocytosis data), we used n=3 biological replicates, with each replicate representing independently harvested bone marrow-derived macrophages from different mice. For Fig. 3h, now included in the supplementary figure 8, the n=10 data points were collected from 3 separate experiments. Here, each 'n' represents a ROI, where we performed measurements on

the entire mitochondrial network within that ROI. Due to the need for high magnification, each ROI contained 3-6 cells.

We treated each ROI data point as an independent measurement to better capture intrinsic biological variability and enhance statistical robustness of our analysis, while also adhering to the principle of reduction in animal use. Similarly, for Fig. 3j, each treatment includes n=9-12 data points that were accumulated from 3 experiments. In this case, each data point, which required the highest magnification, captured 1-5 cells. We treated each cell data point as an independent measurement to further capture the intrinsic biological variability and enhance the statistical robustness of our analysis, again adhering to the reduction of animal use. The details of n identity have been added within the figure legends.

Additional comments:

- To ensure there is no batch effect in the experiment, could the authors perform principal component analysis (PCA) or hierarchical clustering on the full set of data, as the results presented in the scRNA-seq were pooled from five wounds?

We thank the reviewer for their valuable feedback. In our study, the scRNA-seq data were pooled from five mice per treatment group prior to sequencing. This approach precludes the possibility of distinguishing individual mice within the dataset, making it infeasible to perform Principal Component Analysis or hierarchical clustering for replicate-specific batch effect assessment for our current dataset.

We conducted rigorous normalization and data processing, employing algorithms detailed in the 'Sequencing data analysis' section of our Methods. We believe the pooled data accurately represent the cellular components of the wound, providing significant insights.

To establish the effect of treatments on scRNA-seq data, we employed detailed statistical analyses. These included differential clustering (Fig. 1) and differential gene expression analysis across treatments (Supplementary Fig. 2-5, 9-10). To further strengthen the statistical significance of the treatment effect, we have provided additional quality control metrics, as illustrated in panel a below. These include the number of features per treatment (Sham: 20082 genes across 7583 cells; lo-GelMA: 19756 genes across 8008 cells; hi-GelMA: 19905 genes across 6310 cells) and percentage of total gene expression due to mitochondrial DNA (mtDNA), with mtDNA constituting below 5% of the total gene expression of most cells.

Additionally, we conducted a multivariate analysis of variance (MANOVA) on the first 20 principal components inferred from the gene expression data. This analysis revealed a significant effect of the treatments (Sham, lo-GelMA, hi-GelMA) on gene expression profiles (Pillai's trace = 0.15593, Approx F = 92.509, num Df = 40, den Df = 43760, Pr(>F) < 2.2e-16), providing strong evidence that the observed differences are due to treatment effects.

Moreover, when analyzing individual datasets for each treatment separately, they demonstrate similar clustering patterns. Similar clustering patterns observed when analyzing individual datasets for each treatment are illustrated in panel b below, showcasing markers for the main three cell types in our study: *Cd86* (macrophages), *Col1a1* (fibroblasts), and *Csf3r* (neutrophils).

- Could you provide more information on the parameters that were utilized to determine the highlighted area in Figure 1b? Based on the visual data, it appears that the yellow highlighted area may be subjective as there seems to be hair present in the center of the Sham wound.

We thank the reviewer for highlighting the need for this important clarification regarding our methodology. The image originally displayed an ingrown hair within the wound bed, which was visible due to the thin and translucent nature of the wound tissue. Occasionally, hairs can fall into the wound bed during the healing process, with re-epithelialization occurring over them and consequently trapping them inside. To provide a more accurate representation of the typical wound condition, we have replaced this image with another that does not include trapped hairs. In the updated image, the wound edge has been marked at the point where the hair shaft meets the skin surface. Furthermore, we have expanded the details on our measurement technique in the methods section, specifically under the subtitle “Scar size analysis” in lines 635-689.

- In the experiment, was the full excisional model used and was it splinted? There is no information provided in the methodology section. If it was not splinted, how did the authors prevent local skin contraction to ensure that the wound was healed mainly by granulation and re-epithelialization?

We thank the reviewer for highlighting this important aspect of our methodology. We acknowledge that our original manuscript did not sufficiently detail the use of splinting. Consequently, we have updated the Methods section to clarify that traditional external splinting was not utilized. This addition appears in the Methods section under the subtitle “Wounding studies”. However, it is crucial to note that the gels and Tegaderm covering used in our study effectively served as splints, maintaining wound openness, and mitigating local skin contraction. We appreciate the opportunity to clarify this aspect and have included these details in lines 627-628.

- Specify the concentration of recombinant mouse M-CSF used in the methodology section.

We thank the reviewer for this important request. Importantly, we did not use recombinant M-CSF. We used CMG 14–12 cell line expressing recombinant mouse M-CSF and produced conditioned media. We have updated the methods section (line 562) with the following information: The generated conditioned media stock was diluted with complete DMEM at a ratio of 1:10, mixing 50 mL of conditioned media with 500 mL of DMEM. This typically results in concentration of 35,000 U/ml M-CSF (PMID: 10934646).

- The methodology section should include details on how to calculate collagen width, iNOS, Arg+1 expression, and ROS activity.

We thank the reviewer for this important request. In response, we have added the detailed methodology to the Methods section of our manuscript. Information regarding collagen-related measurements is now included under the subtitle “Histology” (lines 647-651), and details pertaining to immunohistochemistry related measurements can be found under the subtitle “Immunofluorescence Staining” (lines 678-680).

- Is the in vitro data a biological replica or a technical replica (with gels being replicated on the same day)?

We thank the reviewer for pointing out that this important information is not clear enough. In response we provide annotation to each “n” in the figure legends using appropriate cells/mice/etc. Of note, all data presentation is biological replicates, gels on different days, cells from different mice. The details of n identity have been added within the figure legends when appropriate.

- The methodology section states that the wound samples were analyzed at 3, 5, 10 and 30 days post-wounding. However, there is no information provided for day 30 in the manuscript.

We thank the reviewer for pointing out that this is not clear enough in the data. Day 30 is used for scarring (Fig. 1b) To clarify we added the title “PWD30” to the Fig. 1b where PWD30 data is presented.

Reviewer#2

The paper discusses the changes in wound healing and related cellular dynamics in mice due to different cross-linking of GelMA, including the relationship between macrophages and fibroblasts, and the effects of hypoxia. It is interesting to note the many single-cell analyses and discussions based on them. However, in this experiment, the difference between hi-GelMA, which is eliminated from the wound area early after transplantation, and lo-GelMA, which remains intact in the wound area, is thought to have a significant effect on the final scar. It is difficult to explain the difference in scar formation when there is a difference in whether or not the foreign material remains in the body. In the case of hi-GelMA, since the blood vessels have not yet invaded the matrix to be eliminated, there is no re-epithelialization on top of this foreign material, and it is difficult to attribute this to a difference in epithelialization due to the difference in cross-linking. The paper should be written from the viewpoint of the difference in biocompatibility and the mechanism of recognition of foreign substances, and from the viewpoint of the relationship between their surface structures and molecular interactions between cells to examine why lo-GelMA and hi-GelMA are more or less likely to be eliminated as foreign substances. We believe that the paper

should be written in the direction to consider the relationship between the surface structure and molecular interaction between cells.

We appreciate Reviewer #2's insightful feedback on the importance of considering biocompatibility and the mechanism of foreign substance recognition in our study. In response, we have made several specific modifications to the manuscript to more explicitly address these aspects, particularly focusing on the differences in cellular interactions between lo-GelMA and hi-GelMA. The key changes are summarized below:

- Biocompatibility and molecular interactions: We updated the Introduction, Results, and Discussion sections to emphasize lo-GelMA's superior biocompatibility and integration into the dermis, contrasting with hi-GelMA's poor biocompatibility and foreign body response (lines 114-135; 475-477; 480-486).
- Differential cellular responses: We revised the text to highlight distinct responses of lo-GelMA and hi-GelMA, linked to their crosslink density (lines 118-120; 123-131). These include adhesion to the hi-GelMA and expression of markers of fusion (new Fig. 5f) and phagocytosis of lo-GelMA (new Fig. 3k and described in more detail below).
- Foreign body recognition: We introduced new section in the Discussion (lines 480-486) focusing on how GelMA hydrogels' surface structures may affect foreign body recognition and its impact on healing.
- Comprehensive conclusion: We updated the Conclusion to reflect the study's findings in terms of biocompatibility, surface properties of biomaterials, and their molecular interactions (lines 597-606).

In addition, we have expanded our analysis to include in vitro phagocytosis studies that compare low and high concentrations of GelMA (lo vs. hi-GelMA) using bone marrow-derived macrophages (Fig. 3k). This new experiment illustrates how GelMA crosslinking affects macrophage uptake. The results not only corroborate our observations from the in vivo scRNA-seq data, which indicated increased phagocytic activity in macrophages in the lo-GelMA condition compared to hi-GelMA, but also suggests a downstream effect of this interaction on macrophage reprogramming and the promotion of wound healing.

Together, these modifications aim to address the reviewer's concerns by providing a more thorough understanding of how GelMA crosslinking affects cellular interactions wound healing, with a particular focus on biocompatibility, foreign substance recognition, and molecular interactions between the hydrogels and the wound environment.

Thank you:

We thank the reviewers for their time and efforts, and we believe the changes made have improved our manuscript. Thank you for considering it for publication in *Nature Communications*.

REVIEWER COMMENTS

Reviewer #1 (Remarks to the Author):

The authors have addressed most of my reviewer comments except one that I think is important given the issue being studied is fibrosis which involves contraction forces.

In my review I asked for clarification about the use of splints to prevent local skin contraction. Again in the context of fibroblast phenotype and healing via granulation tissue formation rather than contraction is important. However, the authors responded by saying they used Tegaderm covering, which “effectively served as splints, maintaining wound openness and mitigating local skin contractions” I'm having a hard time imagining how Tegaderm can achieve this. The authors do not provide further details on the wrapping method to warrant the intended purpose. Given the importance of wound healing modality in the rigor of the results, it is important to demonstrate the lack of contraction in their skin model (not standard way of splinting) and make it very clear in their manuscript results, discussion, and conclusion that this is a limitation of their study so that readers are aware. Whether this contraction significantly changes the fibroblast and macrophage phenotypes and their communication, we do not know.

Reviewer #2 (Remarks to the Author):

I commented on whether or not why lo-GelMA and hi GelMA are more likely to be eliminated as foreign substances and so on, indicating differences in biocompatibility, foreign substance recognition mechanisms, and molecular interactions between surface structures and cells. In response, the author has revised the text and included additional results in Fig. 3k.

However, the additional information in the text is only an interpretation of the results of the present paper and does not prove the mechanism of why the difference in cross-linking caused such a difference in foreign material removal. Therefore, my opinion that my submission to Nature Communications is unsuitable remains unchanged.

REVIEWER COMMENTS

Reviewer #1 (Remarks to the Author):

The authors have addressed all my comments sufficiently. I would have preferred to see the results in a splinted model given fibrosis/contraction is what is being studied, the authors added this limitation to their writeup, so the readers would be aware of this limitation in the study. I am not aware of studies showing that sutures show changes in regeneration with statistical robust analysis.

I would like to continue to state that Tegaderm has not been demonstrated by this study or others to reduce skin contraction and that both statements in lines 506-508 and 581-584, should say that the study was performed without reducing skin contraction at all. Stating that Tegaderm reduces contraction is not factual at this time.

Reviewer #2 (Remarks to the Author):

The authors further elucidated the different M ϕ activities of the M3 and M4 clusters and compared the in vitro dynamics of enhanced phagocytosis of lo-GelMA versus enhanced inflammation and oxidative activation of hi-GelMA.

It is interesting that the new Figure 4 shows the dendrites of highly phagocytic BMDM on lo-GelMA and the flat and rounded morphology characteristic of less phagocytic BMDM on hi-GelMA. However, although the differences in integrins were examined, it is not shown which site on the integrin or other molecule binds to which part of the conformation caused by the difference in lo-GelMA and hi-GelMA cross-linking, resulting in the morphological changes shown in Fig. 4 and the differences in in vivo removal of foreign substances. This has been a consistent question of the reviewer since the initial question and is something we would like to see investigated.

Point by point response to Reviewers

Reviewer 1

- The authors have addressed all my comments sufficiently. I would have preferred to see the results in a splinted model given fibrosis/contraction is what is being studied, the authors added this limitation to their writeup, so the readers would be aware of this limitation in the study. I am not aware of studies showing that sutures show changes in regeneration with statistical robust analysis.

I would like to continue to state that Tegaderm has not been demonstrated by this study or others to reduce skin contraction and that both statements in lines 506-508 and 581-584, should say that the study was performed without reducing skin contraction at all. Stating that Tegaderm reduces contraction is not factual at this time.

We thank the Reviewer for highlighting this important aspect of our methodology and for acknowledging our efforts to address the previous comments. We agree that a splinted wound would provide a better method to reduce skin contraction. We have added a discussion of this important point in the first paragraph of the Discussion section, highlighting the absence of mechanical controls such as splinting in our current study setup. We have also removed the text in original lines 506-508 and 581-584. Together these changes will help readers understand the context and limitations regarding our findings on skin contraction and fibrosis.

Reviewer#2

- The authors further elucidated the different M ϕ activities of the M3 and M4 clusters and compared the in vitro dynamics of enhanced phagocytosis of lo-GelMA versus enhanced inflammation and oxidative activation of hi-GelMA.

It is interesting that the new Figure 4 shows the dendrites of highly phagocytic BMDM on lo-GelMA and the flat and rounded morphology characteristic of less phagocytic BMDM on hi-GelMA. However, although the differences in integrins were examined, it is not shown which site on the integrin or other molecule binds to which part of the conformation caused by the difference in lo-GelMA and hi-GelMA cross-linking, resulting in the morphological changes shown in Fig. 4 and the differences in in vivo removal of foreign substances. This has been a consistent question of the reviewer since the initial question and is something we would like to see investigated.

We appreciate the insightful comments and suggestions. We recognize the importance of understanding specific sites on integrins and other receptors that interact with different GelMA crosslinking densities, which would be crucial for elucidating the morphological changes and functional outcomes in macrophages. While we completely agree with the Reviewer that this represents a significant gap in the field, we respectfully suggest that the detailed mapping of these interactions, on the level of a site on a surface protein, is beyond the scope of the current study. In the *Discussion* section of our manuscript, we now fully acknowledge this gap and highlight it as a key area for future research.

Macrophage receptor-mediated uptake, particularly involving complex materials such as GelMA, encompasses a variety of receptors and potential ligands, making it a dynamic and intricate process. Our prior study (Rowley et al., 2020) illustrated the complexity and potential redundancy in receptor-mediated uptake. In that study, knockdown of LAIR-1, a key collagen receptor, led to increased activity in other receptors including CD18 integrin and SRA1, thereby complicating the isolation of specific receptor activities. Furthermore, the denatured and heterogeneous nature of GelMA is a challenge for epitope identification of this material. For these reasons, we believe isolating specific interactions between macrophage receptors and GelMA would require a comprehensive and focused research project and is beyond the limitations of this study. Our discussion now communicates these scientific and practical challenges, ensuring that readers understand the complexities involved in such an analysis.

In conclusion, we have revised the manuscript to include these discussions, acknowledging both the achievements and the limitations of our study. We hope that these revisions adequately address the concerns raised by the Reviewer and provide a clear path for future research in this important area.

Thank you:

We thank the Reviewers for their time and efforts, and we hope our revisions meet *Nature Communications'* high standards.